# Enhancing Adversarial Robustness Through Robust Information Quantities

## Abstract

It is known that deep neural networks (DNNs) are vulnerable to imperceptible adversarial attacks, and this fact raises concerns about their safety and reliability in real-world applications. In this paper, we aim to boost the robustness of a DNN against white-box adversarial attacks by defining three new information quantities—robust conditional mutual information (CMI), robust separation, and robust normalized CMI (NCMI)—which can serve as robust performance metrics for the DNN. We then utilize these concepts to introduce a novel training method that constrains the robust CMI and increases the robust separation simultaneously. Our experimental results demonstrate that our method consistently enhances model robustness against C&W and AutoAttack on CIFAR and Tiny-ImageNet datasets with and without additional synthetic data. Specifically, it is shown that our approach improves the robust accuracy of a DNN by up to 2.66% on CIFAR datasets and 3.49% on Tiny-ImageNet in the case of PGD attack and 1.70% on CIFAR datasets and 1.63% on Tiny-ImageNet in the case of AutoAttack, in comparison with the state-of-the-art training methods in the literature. Our implementation is publicly available at `https://github.com/ICLR2025-Robust-NCMI/ICLR2025-Robust-NCMI`.

## 1 Introduction

Despite the remarkable success of deep neural networks (DNNs) in computer vision (Krizhevsky et al., 2012; He et al., 2016a) and natural language processing (Vaswani et al., 2017; Devlin et al., 2019), DNNs are found to be vulnerable to adversarial attacks (Szegedy et al., 2013; Goodfellow et al., 2015). These attacks generate adversarial sample instances by adding slight perturbations, imperceptible to human eyes, to the original benign sample instances to deceive the underlying DNN. This raises concern to apply deep learning (DL) models to safety-critical domains like autonomous driving and medical diagnosis (Kurakin et al., 2018; Finlayson et al., 2019).

A simple, yet effective method to train robust DNNs against adversarial attacks is adversarial training (AT) (Goodfellow et al., 2015; Madry et al., 2018). In AT, the model is trained not on benign sample instances, but on adversarial ones. This process is formulated as a min-max optimization problem: the inner maximization focuses on generating adversarial sample instances, while the outer minimization aims to reduce the adversarial loss associated with these attacked sample instances.

Following Madry et al. (2018), a significant body of work has focused on improving model robustness, primarily through four approaches: (i) modifying or adding additional regularization terms to the loss function (Zhang et al., 2019; Wang et al., 2019; Wu et al., 2020), (ii) altering model architecture (Xie et al., 2019), (iii) applying data augmentation techniques (Rebuffi et al., 2021), and (iv) utilizing strategies like early stopping and weight averaging (Rice et al., 2020; Gowal et al., 2020). Even with these enhancements of AT, however, the underlying vulnerability of DNNs against adversarial attacks remains unresolved: data sample instances which are near the decision boundary of DNNs, are in general more susceptible to perturbation and likely to cause misclassification (Zhang et al., 2021).

Now regard a classification DNN as a mathematical mapping from raw data $x \in \mathbb{R}^d$ to a probability distribution $f(x)$ over the set of class labels, predicting an output label $\hat{y}$ with probability $f(x)[\hat{y}]$ in response to input $x$. To mitigate the issue above, we are inspired by the work Yang et al. (2023) and

will analyze in this paper the robustness of a DNN $x \to f(x)$ by examining its information geometry properties in the output probability distribution space. Specifically, for each sample instance $x$, let $x'$ denote its adversarial instance, and refer to $f(x')$ as an adversarial output probability distribution for $x$. In the output probability distribution space, the set of adversarial distributions $f(x')$ for all sample instances $x$ with the same ground truth label $y$ forms a cluster, which is referred to as an adversarial cluster corresponding to $y$ hereafter. Following Yang et al. (2023), the concentration of this adversarial cluster can be measured by the conditional mutual information (CMI) $I(X'; \hat{Y}'|Y = y)$, where $X'$ is an adversarial sample corresponding to the benign sample $X$, $\hat{Y}'$ is the random label predicted by the DNN in response to the input $X'$ with probability $f(X')[\hat{Y}']$, and $Y$ is the ground truth label of $X$. Here $X$, $Y$, $X'$, and $\hat{Y}'$ are all random variables. Averaging over all labels $y$ with respect to the distribution $P_Y(y)$ of $Y$, the CMI $I(X'; \hat{Y}'|Y)$ then measures the average concentration across all adversarial clusters from a given attack method. If for each label $y$, the centroid of the adversarial cluster corresponding to $y$ is close to the one hot probability vector corresponding to $y$, then the smaller $I(X'; \hat{Y}'|Y)$ is, the less likelihood the attack method has to succeed.

In the above, both $I(X'; \hat{Y}'|Y = y)$ and $I(X'; \hat{Y}'|Y)$ depend on the underlying attack method which generates an adversarial instance $x'$ for each $(x, y)$. Since there are many attack methods available and it is unknown which one would be used to attack a learned DNN, we need to go one step further, consider the worst case scenario, and define the robust CMI of the DNN $x \to f(x)$ as

$$I(\epsilon) = \max_{X'} I(X'; \hat{Y}'|Y)$$

where the maximization is taken over all attack methods satisfying $\|x' - x\|_p \leq \epsilon$. In the same spirit, we further extend the concepts of separation and normalized CMI in Yang et al. (2023) to the adversarial case, and define robust separation $\Gamma(\epsilon)$ and robust normalized CMI (NCMI) $\hat{I}(\epsilon)$. Particularly, $\Gamma(\epsilon)$ is the minimum of the inter-class separation between and among adversarial clusters over all attack methods satisfying $\|x' - x\|_p \leq \epsilon$, and $\hat{I}(\epsilon)$ is the maximum of the ratio between $I(X'; \hat{Y}'|Y)$ and the inter-class separation of adversarial clusters over all attack methods satisfying $\|x' - x\|_p \leq \epsilon$.

From the perspective of information theory (or information geometry), the robustness of the DNN $x \to f(x)$ can also be gauged by its robust CMI, robust separation, and robust NCMI. To enhance adversarial robustness, a DNN can then be trained by minimizing its robust CMI while maximizing its robust separation, which is equivalent roughly to minimizing its robust NCMI. After we optimize on such information quantities, the adversarial output probability distributions from the same class would be more concentrated, while those from different classes would be further separated. Hence, less output data points will be near the decision boundaries and considered susceptible to adversarial attacks (see Appendix G.1 for comparison between our method and baseline).

In summary, our contributions are listed as follows:

- We extend the concepts of CMI, separation, and normalized CMI in Yang et al. (2023) to the adversarial case, and introduce three new information quantities, robust CMI $I(\epsilon)$, robust separation $\Gamma(\epsilon)$, and robust NCMI $\hat{I}(\epsilon)$, to gain insights of the intrinsic mapping structure of DNNs in the context of adversarial robustness.

- A new adversarial training framework is presented, in which the robust NCMI can be minimized jointly along with existing training objective functions in AT.

- An alternating learning algorithm is developed to alternatively optimize the weight parameters $\theta$ of the DNN model, and the centroids of adversarial clusters.

- We conduct extensive experiments on CIFAR-$\{10, 100\}$ (Krizhevsky et al., 2009) and Tiny-ImageNet (Le & Yang, 2015) datasets. Our results demonstrate that our proposed learning method indeed boosts adversarial robustness when combined with existing adversarial defense objective functions, both with and without synthetic data.

## 2  RELATED WORK

After Szegedy et al. (2013) showed that DNNs, even though achieving high accuracy on benign data, are vulnerable to imperceptible perturbations, the development of adversarial attacks gained

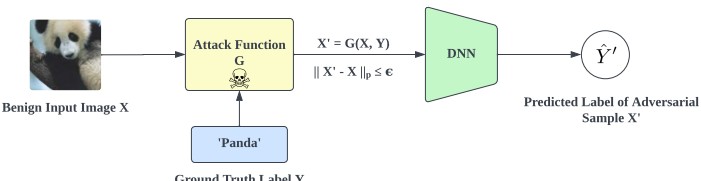

Figure 1: Diagram of generating an adversarial sample instance, where $\hat{Y}'$ is the random label predicted by the DNN in response to the adversarial instance.

attention. Goodfellow et al. (2015) introduced the Fast Gradient Sign Method (FGSM) for crafting adversarial examples with a single gradient step. Later research added randomization and multi-step attack to generate perturbed samples in AT (Tramèr et al., 2018; Kurakin et al., 2018). The success rate of an attack is further improved with projected gradient descent (PGD) by Madry et al. (2018), which iteratively perturbs each benign sample within a bounded neighborhood using a min-max optimization framework to find the corresponding worst-case adversarial example. More recent work even produced stronger attack by crafting perturbed samples in an adaptive and parametric-free manner but came with a drawback of higher computational cost (Croce & Hein, 2020). In general, an adversarial sample $x'$ is crafted as following:

$$x' = G(x, y) = x + \underset{\|\delta\|_p \leq \epsilon}{\arg\max} \mathcal{L}(x + \delta, y), \tag{1}$$

where $G(x, y)$ represents the attack function to generate adversarial example $x'$ given benign input $x$ and its corresponding ground truth label $y$, $\delta$ is the perturbation to be added to $x$ to generate adversarial sample $x'$, $\|\cdot\|_p$ indicates the $L_p$-norm of a vector, $\epsilon$ denotes the maximum perturbation allowed to generate an adversarial sample, and $\mathcal{L}(\cdot, \cdot)$ stands for the objective function, which is typically cross entropy (CE) loss in previous works.

As an AT defense mechanism, vanilla AT (Madry et al., 2018) showed great success by training a DNN with samples perturbed by PGD attack. Subsequent studies utilized its robust optimization to generate adversarial samples and employed the min-max framework to minimize error rates, while also refining loss formulations to train DNNs for enhanced robustness (Kannan et al., 2018; Zhang et al., 2019; Wang et al., 2019; Wu et al., 2020). Other approaches focused on improving adversarial performance by studying the margin between each adversarial sample and the decision boundaries (Ding et al., 2020; Rade & Moosavi-Dezfooli, 2022; Xu et al., 2023b) or by assigning greater weights to vulnerable examples near such boundaries (Liu et al., 2021; Zhang et al., 2021). Despite all above methods achieved good robust performance, they considered each attacked data point individually, and overlooked to view adversarial samples with the same label as a cluster.

In our approach, we argue that robustness can be further enhanced by constraining adversarial examples from a cluster perspective, ensuring that the worst-case perturbed samples within a class remain as close to the centroid of their corresponding adversarial cluster as possible. Furthermore, since previous research demonstrated that additional labelled or unlabelled data (Schmidt et al., 2018; Carmon et al., 2019; Uesato et al., 2019) as well as synthetic data created by generative models (Gowal et al., 2021; Rebuffi et al., 2021; Sehwag et al., 2022; Wang et al., 2023) can largely boost robust performance, we evaluate our method on datasets both with and without synthetic data, and present our methodology and results in the following sections.

## 3 NEW INFORMATION QUANTITIES FOR ROBUST PERFORMANCE OF DNNS

In this section, we extend the concepts of CMI, separation, and normalized CMI in Yang et al. (2023) to the adversarial case, and introduce three new information quantities, robust CMI $I(\epsilon)$, robust separation $\Gamma(\epsilon)$, and robust NCMI $\hat{I}(\epsilon)$, to gauge the robust performance of DNNs. We begin with notation to be used throughout the paper.

### 3.1 NOTATION

For a positive integer $K$, let $[K] = \{1, \ldots, K\}$ denote a set of integers starting from 1 to $K$. Assume there are $C$ class labels, with $[C]$ indicating the set of all such labels. Let $\mathcal{P}([C])$ denote the set of

all probability distributions over $[C]$. We also use $s[i]$ to represent the $i$-th entry of a probability distribution $s$. Given any two probability distributions $s_1, s_2 \in \mathcal{P}([C])$, we define CE of $s_1$ and $s_2$ as:

$$H(s_1, s_2) = \sum_{i=1}^{C} -s_1[i] \ln s_2[i] \tag{2}$$

and the Kullback-Leibler (KL) divergence between $s_1$ and $s_2$ as:

$$D(s_1 \| s_2) = \sum_{i=1}^{C} s_1[i] \ln \frac{s_1[i]}{s_2[i]}. \tag{3}$$

For any random vector $(X, Y)$, we use $P_{X,Y}(x, y)$ or $P(x, y)$ to denote its joint distribution, and $P_X(x)$ and $P_Y(y)$ (or simply $P(x)$ and $P(y)$) to denote the marginal distributions of $X$ and $Y$, respectively. The conditional distribution of $Y$ given $X = x$ is written as $P_{Y|X}(\cdot|x)$, and that of $X$ given $Y = y$ is denoted by $P_{X|Y}(\cdot|y)$. Additionally, denote $\mathbb{E}[\cdot]$ as expectation, and $\mathbb{E}_X[\cdot]$ as expectation with respect to random variable $X$. To clarify, we use the term 'budget $\epsilon$' to represent the maximum perturbation allowed to generate an adversarial instance $x'$ within the $L_p$-norm ball centered at the benign sample instance $x$ with a radius of $\epsilon$. Furthermore, we use $G(x, y)$ to represent an attack function which crafts, for each $(x, y)$, an adversarial instance $x'$ satisfying $\|x' - x\|_p \leq \epsilon$.

### 3.2 ROBUST ERROR RATE

Given a DNN $x \to f(x)$, one metric used to measure its robust performance is its robust error rate. Let $(X, Y)$ be a random sample the distribution of which governs either a training dataset or a testing dataset, where $Y$ is the ground truth label of $X$. Receiving the benign sample $X$, the DNN outputs a predicted random label $\hat{Y}$ with probability $f(X)[\hat{Y}]$ given $X$. The benign error rate of the DNN for $(X, Y)$ is equal to

$$\xi = \Pr(\hat{Y} \neq Y). \tag{4}$$

With reference to Figure 1, once $X$ is attacked and replaced by an adversarial sample $X'$, the DNN outputs a predicted random label $\hat{Y}'$ with probability $f(X')[\hat{Y}']$ given $X'$. The corresponding adversarial error rate of the DNN for $(X, Y)$ is equal to

$$\xi' = \Pr(\hat{Y}' \neq Y). \tag{5}$$

Since we need to consider all possible attack functions $G$ satisfying $\|x' - x\|_p \leq \epsilon$, define the robust error rate of the DNN as follows

$$\xi_{\mathrm{r}}(\epsilon) = \max_{G: \|x'-x\|_p \leq \epsilon} \Pr(\hat{Y}' \neq Y). \tag{6}$$

Then we have the following theorem.

**Theorem 1** *For any DNN $x \to f(x)$, any $(X, Y)$, and any $\epsilon \geq 0$,*

$$\xi_r(\epsilon) \leq \max_{G: \|x'-x\|_p \leq \epsilon} \mathbb{E}_{X'}[H(P_{Y|X'}(\cdot|X'), f(X'))]$$
$$= \sum_{(x,y)} P(x, y) \left[ \max_{x': \|x'-x\|_p \leq \epsilon} \{ -\ln f(x')[y] \} \right]. \tag{7}$$

Theorem 1 will be proved in Appendix A. As Theorem 1 suggests, to minimize $\xi_{\mathrm{r}}(\epsilon)$, one can train a DNN model by minimizing the upper bound in equation 7, which is exactly what vanilla AT (Madry et al., 2018) and MART (Wang et al., 2019) do essentially.

### 3.3 EXTENSION OF CMI, SEPARATION, AND NCMI TO THE ADVERSARIAL CASE

When benign sample instances $x$ are fed into the DNN, the set of output distributions $f(x)$ for all benign instances $x$ with the same ground truth label $y$ forms a benign cluster in $\mathcal{P}([C])$ corresponding to $y$. As shown in Yang et al. (2023), the centroid of this benign cluster is

$$s_y = P_{\hat{Y}|Y}(\cdot|Y = y) = \sum_x P_{X|Y}(x|y) f(x) = \mathbb{E}_{X|Y}[f(X)|Y = y]; \tag{8}$$

and the concentration of this benign cluster is measured by the CMI $I(X; \hat{Y}|Y = y)$

$$I(X; \hat{Y}|Y = y) = \sum_x P_{X|Y}(x|y) \left[ \sum_{i=1}^C P_{\hat{Y}|XY}(\hat{Y} = i|x, y) \ln \frac{P_{\hat{Y}|XY}(\hat{Y} = i|x, y)}{P_{\hat{Y}|Y}(i|y)} \right] \quad (9)$$

$$= \mathbb{E}_{X|Y} \left[ \left( \sum_{i=1}^C f(X)[i] \ln \frac{f(X)[i]}{P_{\hat{Y}|Y}(i|y)} \right) \mid Y = y \right] \quad (10)$$

$$= \mathbb{E}_{X|Y} \left[ D\left(f(X)\|s_y\right) \mid Y = y \right]. \quad (11)$$

Averaging over all labels $y$, the CMI $I(X; \hat{Y}|Y)$

$$I(X; \hat{Y}|Y) = \sum_{y \in [C]} P_Y(y) \cdot I(X; \hat{Y}|Y = y) = \mathbb{E}_{XY} \left[ D\left(f(X)\|s_Y\right) \right] \quad (12)$$

then measures the average concentration across all benign clusters.

In Yang et al. (2023), the separation between and among all benign clusters is defined as

$$\Gamma = \mathbb{E} \left[ I_{\{Y \neq V\}} H(f(X), f(U)) \right] \quad (13)$$

where $(U, V)$ is another pair of random variables independent of $(X, Y)$, and having the same joint distribution as that of $(X, Y)$, and $I_{\{Y \neq V\}}$ denotes the indicator function of the event $\{Y \neq V\}$. The NCMI for benign clusters is then defined as

$$\hat{I}(X; \hat{Y}|Y) = \frac{I(X; \hat{Y}|Y)}{\Gamma}. \quad (14)$$

When $P(x, y)$ is unknown and approximated by the empirical distribution of a dataset $\mathcal{D} = \{(x_j, y_j)\}_{j=1}^n$, the above information quantities are computed according to their respective sample means:

$$s_y = \frac{1}{|\mathcal{D}_y|} \sum_{j \in \mathcal{D}_y} f(x_j) \quad (15)$$

$$I(X; \hat{Y}|Y = y) = \frac{1}{|\mathcal{D}_y|} \sum_{j \in \mathcal{D}_y} D\left(f(x_j)\|s_y\right) \quad (16)$$

$$I(X; \hat{Y}|Y) = \frac{1}{n} \sum_{j=1}^n D\left(f(x_j)\|s_{y_j}\right) \quad (17)$$

$$\Gamma = \frac{1}{n^2} \sum_{j=1}^n \sum_{k=1}^n I_{\{y_j \neq y_k\}} H(f(x_j), f(x_k)), \quad (18)$$

where $\mathcal{D}_y = \{j \,|1 \leq j \leq n, y_j = y\}$ and $|\mathcal{D}_y|$ denotes the cardinality of $\mathcal{D}_y$.

With reference to Figure 1 again, when benign sample instances $x$ are attacked and replaced by respective adversarial sample instances $x' = G(x, y)$, we have adversarial clusters in $\mathcal{P}([C])$, one per label $y$. The above information quantities can be carried over to characterize the information geometry properties of these adversarial clusters. Simply replacing $(X, \hat{Y}, U)$ by $(X', \hat{Y}', U')$ in equation 11 to equation 14, where

$$X' = G(X, Y) \text{ and } U' = G(U, V), \quad (19)$$

we get the corresponding adversarial counterparts:

$$I(X'; \hat{Y}'|Y = y) = \mathbb{E}_{X'|Y} \left[ D\left(f(X')\|s_y'\right) \mid Y = y \right] = \mathbb{E}_{X|Y} \left[ D\left(f(G(X, Y))\|s_y'\right) \mid Y = y \right] \quad (20)$$

$$I(X'; \hat{Y}'|Y) = \mathbb{E}_{X'Y} \left[ D\left(f(X')\|s_Y'\right) \right] = \mathbb{E}_{XY} \left[ D\left(f(G(X, Y))\|s_Y'\right) \right] \quad (21)$$

$$\Gamma' = \mathbb{E} \left[ I_{\{Y \neq V\}} H(f(X'), f(U')) \right] = \mathbb{E} \left[ I_{\{Y \neq V\}} H(f(G(X, Y)), f(G(U, V))) \right] \quad (22)$$

$$\hat{I}(X'; \hat{Y}'|Y) = \frac{I(X'; \hat{Y}'|Y)}{\Gamma'}, \quad (23)$$

where

$$s'_y = P_{\hat{Y}'|Y}(\cdot|Y=y) = \mathbb{E}_{X|Y}\left[f(G(X,Y))|Y=y\right]. \tag{24}$$

Note that given the attack function $G$, since $P_{\hat{Y}'|XY}(\hat{Y}'=i|x,y) = P_{\hat{Y}'|X'Y}(\hat{Y}'=i|x',y) = f(x')[i]$, it follows that

$$I(X;\hat{Y}'|Y=y) = I(X';\hat{Y}'|Y=y) \text{ and } I(X;\hat{Y}'|Y) = I(X';\hat{Y}'|Y).$$

Hereafter, we will use $I(X;\hat{Y}'|Y)$ and $I(X';\hat{Y}'|Y)$ interchangeably. Again, when $P(x,y)$ is unknown, all quantities in equation 20 to equation 24 can be computed according to their respective sample means over a dataset $\mathcal{D} = \{(x_j,y_j)\}_{j=1}^n$. In other words, equation 15 to equation 18 remain valid if $(s_y, X, \hat{Y}, \Gamma, x_j, x_k)$ is replaced by $(s'_y, X', \hat{Y}', \Gamma', x'_j, x'_k)$, where $x'_j = G(x_j, y_j)$.

Given a DNN, let $G$ be the attack function given by AutoAttack (Croce & Hein, 2020). We have trained PreAct-ResNet-18 (He et al., 2016b), using various AT methods including the standard CE method without adversarial instances (denoted as 'CE'), on CIFAR-10, and then evaluated the robust accuracy of these learned DNNs against AutoAttack. Fig. 2 illustrates the robust accuracy of these learned DNNs against AutoAttack vs their adversarial NCMI corresponding to $G$. It is clear from Fig. 2 that across these different learned DNNs, the robust accuracy tends to be inversely proportional to the corresponding adversarial NCMI, which is consistent with the observation made by Yang et al. (2023) in the benign case.

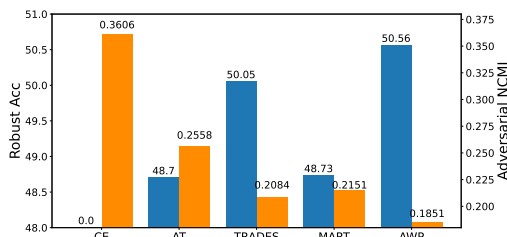

Figure 2: Robust accuracy of different learned models against AutoAttack vs the corresponding adversarial NCMI on CIFAR-10, where blue and orange bars represent robust accuracy (left axis) and adversarial NCMI (right axis), respectively, with corresponding specific values shown on top of each bar.

### 3.4 ROBUST CMI, SEPARATION, & NCMI

Note that all adversarial information quantities in equation 20 to equation 24 depend on the underlying attack function $G$. Since given a DNN, there are many attack functions available and it is unknown which one would be used to attack the DNN, we need to consider the worst case. To this end, we have the following definition with reference to Figure 1.

**Definition 1** *Given a DNN $x \to f_\theta(x)$ with its weight parameters $\theta$, define its robust CMI, separation, and NCMI over a random sample $(X, Y)$ respectively as follows:*

$$I(X, Y, \theta, \epsilon) = \max_{G:\|x'-x\|_p \leq \epsilon} I(X;\hat{Y}'|Y) \tag{25}$$

$$\Gamma(X, Y, \theta, \epsilon) = \min_{G:\|x'-x\|_p \leq \epsilon} \mathbb{E}\left[I_{\{Y\neq V\}}H(f_\theta(G(X,Y)), f_\theta(G(U,V)))\right] \tag{26}$$

$$\hat{I}(X, Y, \theta, \epsilon) = \max_{G:\|x'-x\|_p \leq \epsilon} \frac{I(X;\hat{Y}'|Y)}{\mathbb{E}\left[I_{\{Y\neq V\}}H(f_\theta(G(X,Y)), f_\theta(G(U,V)))\right]}. \tag{27}$$

*Whenever $(X, Y, \theta)$ is clear from the context, simply write $I(X, Y, \theta, \epsilon)$, $\Gamma(X, Y, \theta, \epsilon)$, and $\hat{I}(X, Y, \theta, \epsilon)$ as $I(\epsilon)$, $\Gamma(\epsilon)$, and $\hat{I}(\epsilon)$, respectively.*

Given $(X, Y, \theta)$, $\Gamma(\epsilon)$ can be computed via the standard gradient decent method through backpropagation. By introducing a dummy conditional "backward channel" $q(x|i,y)$ from $\hat{Y}'$ to $X$ given $Y$, $I(\epsilon)$ and $\hat{I}(\epsilon)$ can be computed via an alternating method, as implied by the following theorem.

**Theorem 2** *For any $i, y \in [C]$, let $q(\cdot|i,y)$ denote a dummy conditional distribution in the space of $X$, which is assumed to be discrete for simplicity. Then $I(\epsilon)$ and $\hat{I}(\epsilon)$ can be computed by solving*

*the following double maximization problems, respectively:*

$$I(\epsilon) = \max_{\{q(x|i,y)\}} \max_{G:\|x'-x\|_p \le \epsilon} \sum_{x,y} P(x,y) \left[ \sum_{i=1}^{C} f_\theta(G(x,y))[i] \ln q(x|i,y) - \ln P_{X|Y}(x|y) \right] \quad (28)$$

$$= \max_{\{q(x|i,y)\}} \sum_{x,y} P(x,y) \left[ \max_{x':\|x'-x\|_p \le \epsilon} \sum_{i=1}^{C} f_\theta(x')[i] \ln q(x|i,y) - \ln P_{X|Y}(x|y) \right] \quad (29)$$

$$\hat{I}(\epsilon) = \max_{\{q(x|i,y)\}} \max_{G:\|x'-x\|_p \le \epsilon} \frac{\sum_{x,y} P(x,y) \left[ \sum_{i=1}^{C} f_\theta(G(x,y))[i] \ln q(x|i,y) - \ln P_{X|Y}(x|y) \right]}{\mathbb{E}\left[ I_{\{Y \ne V\}} H(f_\theta(G(X,Y)), f_\theta(G(U,V))) \right]} \quad (30)$$

*where given G, the outer maximization above is achieved when*

$$q(x|i,y) = \frac{P_{X|Y}(x|y) f_\theta(x')[i]}{P_{\hat{Y}'|Y}(i|y)}. \quad (31)$$

Theorem 2 will be proved in Appendix B. Given $\{q(x|i,y)\}$, the inner maximization in equation 28 to equation 30 can now be computed via the standard gradient decent method through backpropagation. Thus we can compute $I(\epsilon)$ and $\hat{I}(\epsilon)$ iteratively by alternatively optimizing $\{q(x|i,y)\}$ given $G$, and $G$ given $\{q(x|i,y)\}$. Illustrative examples will be given in Appendix B as well.

## 4 $\hat{I}(\epsilon)$ CONSTRAINED ADVERSARIAL TRAINING

Given a DNN architecture, in order to enhance adversarial robustness, it follows from the above discussions that a desirable learning algorithm should generate $\theta$ so that both $\hat{I}(X,Y,\theta,\epsilon)$ and a conventional AT loss are small. Let $\mathcal{L}(X,\hat{X},Y)$ denote a conventional AT loss such as one of those in TRADES (Zhang et al., 2019) and MART (Wang et al., 2019), where $\hat{X} = G_1(X,Y)$ is a perturbed sample instance given by the attack function $G_1$ in the conventional AT loss. Then instead of training the DNN by solving

$$\min_\theta \mathbb{E}[\max_{G_1:\|\hat{x}-x\|_p \le \epsilon} \mathcal{L}(X,\hat{X},Y)], \quad (32)$$

one tries to solve the following optimization problem

$$\min_\theta \left\{ \mathbb{E}[\max_{G_1:\|\hat{x}-x\|_p \le \epsilon} \mathcal{L}(X,\hat{X},Y)] + \lambda \hat{I}(X,Y,\theta,\epsilon) \right\} \quad (33)$$

which, due to equation 27, equation 21, & equation 22, is equivalent to

$$\min_\theta \left\{ \mathbb{E}[\max_{G_1:\|\hat{x}-x\|_p \le \epsilon} \mathcal{L}(X,\hat{X},Y)] + \max_{G:\|x'-x\|_p \le \epsilon} \left[ \alpha \cdot I(X';\hat{Y}'|Y) - \beta \cdot \Gamma' \right] \right\} \quad (34)$$

$$= \min_\theta \left\{ \mathbb{E}[\max_{G_1:\|\hat{x}-x\|_p \le \epsilon} \mathcal{L}(X,\hat{X},Y)] + \max_{G:\|x'-x\|_p \le \epsilon} \left[ \alpha \mathbb{E}[D(f_\theta(G(X,Y))\|s_Y')] \right. \right.$$

$$\left. \left. - \beta \mathbb{E}[I_{\{Y \ne V\}} H(f_\theta(G(X,Y)), f_\theta(G(U,V)))] \right] \right\} \quad (35)$$

$$= \min_\theta \max_{G_1} \max_G \left\{ \mathbb{E}[\mathcal{L}(X, G_1(X,Y), Y)] + \alpha \mathbb{E}[D(f_\theta(G(X,Y))\|s_Y')] \right.$$

$$\left. - \beta \mathbb{E}[I_{\{Y \ne V\}} H(f_\theta(G(X,Y)), f_\theta(G(U,V)))] \right\} \quad (36)$$

where in the above, the first maximization is over all $G_1$ satisfying $\|\hat{x} - x\|_p \le \epsilon$, and the second maximization is over all $G$ satisfying $\|x' - x\|_p \le \epsilon$.

Due to the dependency of $s_y'$ on the entire adversarial cluster corresponding to $y$, the second term of the objective function in equation 36 is not amenable to parallel computation via GPU. To overcome this difficulty, we follow Yang et al. (2023)) and introduce a dummy adversarial centroid distribution $Q_y'$ for each label $y$. Then we have the following result, which is proved Appendix C.

**Theorem 3** *For any $\epsilon \geq 0$,*

$$
\min_{\theta} \left\{ \mathbb{E}[\max_{G_1 : \|\hat{x} - x\|_p \leq \epsilon} \mathcal{L}(X, \hat{X}, Y)] + \max_{G : \|x' - x\|_p \leq \epsilon} \left[ \alpha \cdot I(X'; \hat{Y}'|Y) - \beta \cdot \Gamma' \right] \right\}
$$

$$
\leq \min_{\theta} \min_{\{Q'_c\}_{c \in [C]}} \max_{G_1, G} \left\{ \mathbb{E}[\mathcal{L}(X, G_1(X, Y), Y)] + \alpha \mathbb{E}[D(f_\theta(G(X, Y)) \| Q'_Y)] \right.
$$

$$
\left. - \beta \mathbb{E}[I_{\{Y \neq V\}} H(f_\theta(G(X, Y)), f_\theta(G(U, V)))] \right\} \tag{37}
$$

*where given $(\theta, G_1, G)$, the minimization over $\{Q'_c\}_{c \in [C]}$ is achieved when*

$$
Q'_y = s'_y = \mathbb{E}_{X|Y} \left[ f_\theta(G(X, Y)) | Y = y \right]. \tag{38}
$$

Given $(\theta, \{Q'_c\}_{c \in [C]}, G_1, G)$, the objective function on the right side of equation 37 is now sample additive, and becomes

$$
\mathcal{J}_\mathcal{B}(\theta, \{Q'_c\}_{c \in [C]}, G_1, G) = \left\{ \frac{1}{|\mathcal{B}|} \sum_{(x, y) \in \mathcal{B}} \mathcal{L}(x, G_1(x, y), y) + \frac{1}{|\mathcal{B}|} \mathcal{L}_\mathcal{B}(\theta, \{Q'_c\}_{c \in [C]}, G) \right\} \tag{39}
$$

when $P(x, y)$ is unknown and approximated by the empirical distribution of a mini-batch $\mathcal{B}$, where

$$
\mathcal{L}_\mathcal{B}(\theta, \{Q'_c\}_{c \in [C]}, G) = \sum_{(x, y) \in \mathcal{B}} \left[ \alpha D(f_\theta(G(x, y)) \| Q'_y) \right.
$$

$$
\left. - \frac{\beta}{|\mathcal{B}|} \sum_{(u, v) \in \mathcal{B}} I_{\{y \neq v\}} H\left( f_\theta(G(x, y)), f_\theta(G(u, v)) \right) \right]. \tag{40}
$$

In $\hat{I}(\epsilon)$ constrained AT, we solve the optimization problem on the right side of equation 37 or

$$
\min_{\theta} \min_{\{Q'_c\}_{c \in [C]}} \max_{G_1, G} \mathcal{J}_\mathcal{B}(\theta, \{Q'_c\}_{c \in [C]}, G_1, G) \tag{41}
$$

when $P(x, y)$ is unknown and approximated by the empirical distribution of a mini-batch $\mathcal{B}$.

Given $(\theta, \{Q'_c\}_{c \in [C]})$, the optimal $G_1^*$ in equation 41 can be found by computing, for each $(x, y) \in \mathcal{B}$, a perturbed instance $\hat{x}$ individually. However, the optimal $G^*$ in equation 41 has to be found by computing all adversarial instances $x'$ for all $(x, y) \in \mathcal{B}$ simultaneously all at once due to the interconnection in the adversarial separation $\Gamma'$. Once $G_1^*$ and $G^*$ are determined, $\theta$ and $\{Q'_c\}_{c \in [C]}$ can be optimized alternatingly: (1) fix $\{Q'_c\}_{c \in [C]}$ and then perform back-propagation to update $\theta$; (2) fix the optimized $\theta$ and then update $\{Q'_c\}_{c \in [C]}$ according to equation 38, but in a weighted manner. The pseudo-code of this alternating learning algorithm is presented in Appendix D as Algorithm 1, where it is assumed that there are a sufficient number of instances in each adversarial cluster within a mini-batch. When this assumption is not valid, we can update $\{Q'_c\}_{c \in [C]}$ at the end of each epoch using adversarial instances for the entire training set. This relaxed alternating learning algorithm is shown in Appendix E.

## 5 EXPERIMENTAL RESULTS

To validate the efficacy of our proposed $\hat{I}(\epsilon)$ constrained AT, we integrated it with several adversarial loss functions, including vanilla AT (Madry et al., 2018), TRADES (Zhang et al., 2019), MART (Wang et al., 2019), and AWP (Wu et al., 2020), and considered the standard version of such AT methods as baselines. We conducted extensive experiments[1] on CIFAR-10, 100 (Krizhevsky et al., 2009), Tiny-ImageNet (Le & Yang, 2015), and CIFAR-10, 100 augmented with 1 million synthetic images. Our results show that incorporating our method improves the performance of these loss functions on both original and synthetic-augmented datasets.

---

[1]Experiments on CIFAR-100 without synthetic data and Tiny-ImageNet followed the algorithm in Appendix E due to the small number of samples from each class per mini-batch, which may cause inaccurate centroid estimation using Alg. 1. All other experiments in this section follow Alg. 1.

In all of our experiments, training adversarial samples were generated using PGD with an $L_\infty$-norm budget of $\epsilon = 8/255$, a step size of $2/255$, and optimized over 10 steps. To evaluate robustness, we applied multiple white-box attacks, including PGD-40 (Madry et al., 2018), C&W attack (Carlini & Wagner, 2017) under $L_\infty$-norm (both with 40 steps), and AutoAttack (Croce & Hein, 2020), to the model checkpoint with the highest validation accuracy. We used default hyper-parameter settings for all baseline AT methods and selected the hyper-parameters $\alpha$ and $\beta$ for our $\hat{I}(\epsilon)$ constrained AT from the set of $\{1 \times 10^{-3}, \ldots, 1 \times 10^0, 2, 5\}$.

## 5.1 RESULTS ON TINY-IMAGENET

To demonstrate the effectiveness of our method, we combined $\hat{I}(\epsilon)$ constrained AT with vanilla AT and TRADES on the Tiny-ImageNet dataset using PreAct-ResNet-$\{18, 34\}$ architectures. Models were trained for 100 epochs with a mini-batch size of 256, $L_2$ regularization ($5 \times 10^{-4}$), and a cosine annealing learning rate scheduling (max 0.1). Weight averaging (WA) (Izmailov et al., 2018) was applied with a decay

Table 1: Experimental results on Tiny-ImageNet.

| Model | Method | Clean | PGD | C&W$_\infty$ | AA |
|-------|--------|-------|-----|------|-----|
| PR-18 | TRADES | **47.13** | 23.12 | 18.47 | 17.59 |
| | +Ours | 46.95 | **23.30** | **18.91** | **18.10** |
| | Vanilla AT | **49.08** | 23.15 | 20.79 | 18.78 |
| | +Ours | 48.43 | **26.29** | **21.41** | **19.72** |
| PR-34 | TRADES | **49.17** | 24.69 | 20.59 | 19.86 |
| | +Ours | 48.56 | **25.09** | **20.94** | **20.37** |
| | Vanilla AT | 50.13 | 24.19 | 22.16 | 20.06 |
| | +Ours | **50.99** | **27.68** | **23.87** | **21.69** |

factor of $\tau = 0.995$, as adopting WA can smooth the loss landscape, thereby reducing robust generalization gap and leading to enhanced performance (Gowal et al., 2020; Chen et al., 2021). In our approach, the baseline adversarial loss function was used until the 5th epoch to maintain stability, and the objective function in equation 41 was applied afterwards, since adversarial clusters are not well-formed in the early stage of training. In Table 1, we reported the accuracy on benign samples, robust accruacy against PGD, C&W and AutoAttack, and denoted such metrics as 'Clean', 'PGD', 'C&W$_\infty$' and 'AA', respectively. As shown, our method exhibits enhanced robustness comparing with those without constraining on $\hat{I}(\epsilon)$, and the robust accuracy gap across all attacks increases as the model size grows. In addition, we observed that the gain of robust accuracy against PGD attack is more than 3% when incorporating our approach with vanilla AT for both models, and reaches a maximum gain of 1.63% against AutoAttack.

## 5.2 RESULTS ON CIFAR

In our experiments on the original CIFAR datasets without synthetic data, we used PreAct-ResNet and Wide ResNet (WRN) (Zagoruyko & Komodakis, 2016) with ReLU activation, both common architectures in AT. Models were trained for 200 epochs with a mini-batch size of 128 and a multi-step decay learning rate schedule, starting at 0.1 and reduced by a factor of 0.1 at the 100th and 150th epoch. We used stochastic gradient descent (SGD) with momentum of 0.9, $L_2$ regularization factor of $5 \times 10^{-4}$, and a WA factor $\tau = 0.999$. The adversarial centroids $Q'_Y$ were updated per mini-batch using WA with a factor $\tau_Q = 0.999$.

For experiments with additional data, we used 1 million labeled synthetic images from Wang et al. (2023), and replaced ReLU with Swish activation (Ramachandran et al., 2017) for better smoothness (Xie et al., 2020; Gowal et al., 2020). Models were trained for 400 epochs with WA ($\tau = 0.995$) and centroids were updated with a WA factor $\tau_Q = 0.99$. We applied cosine annealing for the learning rate scheduling, with a maximum value of 0.2. The original-to-synthetic data ratio was set to 3:7, and mini-batch size is set to 512. Other settings are identical to those of the original CIFAR datasets.

We demonstrate our experimental results on original CIFAR-$\{10, 100\}$ in Table 2, where employing our method alongside existing adversarial loss functions consistently outperforms training solely with those loss functions without constraining on $\hat{I}(\epsilon)$ across all methods when evaluated under AutoAttack. In most settings, our method slightly sacrifices benign accuracy for better robustness, while we observed that both benign and robust accuracy against AutoAttack increase for most models trained by vanilla AT and TRADES with constraining on $\hat{I}(\epsilon)$ comparing with its counterparts without it. We also present the loss curves and a 2-dimensional probability simplex indicating the output distributions with different constraining levels on robust NCMI in Appendix G.

Moreover, we reported experimental results over CIFAR datasets with additional 1 million synthetic data in Table 2 as well. Similar to those trained solely on original datasets, $\hat{I}(\epsilon)$ constrained

Table 2: Experimental results on CIFAR-{10, 100} datasets, both with and without synthetic data, are averaged over 3 runs. 'Clean' refers to accuracy on benign samples, 'C&W$_\infty$' indicates robust accuracy against C&W attack using PGD-40 under $L_\infty$-norm, and 'AA' represents robust accuracy against AutoAttack. '+Ours' indicates the use of our $\hat{I}(\epsilon)$ constrained AT. Results for original datasets are listed under 'No Synthetic Data', while those with additional synthetic data are under '1M Synthetic Data'. WRN-34-10 was used only for original datasets (in red cells), and WRN-28-10 only for datasets with synthetic data (in blue cells). Better results are in **bold**.

| Dataset | Model | Method | No Synthetic Data | | | | 1M Synthetic Data | | | |
|---|---|---|---|---|---|---|---|---|---|---|
| | | | Clean | PGD | C&W$_\infty$ | AA | Clean | PGD | C&W$_\infty$ | AA |
| CIFAR-10 | PreAct-RN18 | Vanilla AT | 79.35 | 52.51 | 50.76 | 48.70 | **90.08** | 57.65 | 57.24 | 54.91 |
| | | +Ours | **81.43** | **53.04** | **52.23** | **49.47** | 89.88 | **60.31** | **59.30** | **56.61** |
| | | TRADES | 82.21 | 53.81 | 51.25 | 50.05 | **88.48** | 61.16 | 58.87 | 58.13 |
| | | +Ours | **82.26** | **54.12** | **51.65** | **50.70** | 87.90 | **61.51** | **59.80** | **58.52** |
| | | MART | **79.81** | **54.60** | 50.63 | 48.73 | 86.99 | **58.46** | 54.68 | 52.86 |
| | | +Ours | 79.40 | 54.41 | **51.25** | **49.61** | **87.82** | 57.54 | **56.05** | **53.92** |
| | | TRADES-AWP | 81.52 | 54.40 | 51.63 | 50.56 | 87.69 | 60.77 | 58.20 | 57.54 |
| | | +Ours | **81.73** | **54.64** | **52.15** | **51.07** | **87.73** | **61.18** | **59.07** | **58.25** |
| | WRN | Vanilla AT | 85.06 | 56.23 | 56.00 | 53.58 | **92.69** | 63.47 | 64.06 | 61.47 |
| | | +Ours | **85.58** | **57.25** | **56.20** | **54.22** | 92.05 | **65.56** | **64.26** | **61.71** |
| | | TRADES | **85.54** | 57.19 | 56.26 | 55.03 | 90.26 | 64.95 | 63.94 | 62.73 |
| | | +Ours | 85.24 | **57.56** | **56.60** | **55.27** | **90.75** | **65.45** | **64.77** | **63.27** |
| | | MART | **84.15** | **57.64** | 55.40 | 53.55 | **91.91** | **64.71** | 63.26 | 61.21 |
| | | +Ours | 83.99 | 57.20 | **55.92** | **53.77** | 91.84 | 63.74 | **63.56** | **61.75** |
| | | TRADES-AWP | **85.14** | 58.02 | 56.66 | 55.35 | **90.44** | 65.56 | 63.93 | 63.17 |
| | | +Ours | 84.67 | **58.25** | **56.81** | **55.60** | 90.37 | **66.15** | **64.53** | **63.41** |
| CIFAR-100 | PreAct-RN18 | Vanilla AT | 52.46 | 29.08 | 26.76 | 24.90 | **66.95** | 32.23 | 32.32 | 29.76 |
| | | +Ours | **54.13** | **31.50** | **27.59** | **25.70** | 65.93 | **33.51** | **33.45** | **30.88** |
| | | TRADES | 55.48 | 28.55 | 25.57 | 24.36 | **63.83** | 36.08 | 32.84 | 31.92 |
| | | +Ours | **57.95** | **29.92** | **26.00** | **24.76** | 63.73 | **36.32** | **33.08** | **32.45** |
| | | MART | **52.10** | 30.34 | 26.48 | 25.00 | 62.44 | 35.10 | 31.62 | 29.87 |
| | | +Ours | 51.46 | **30.70** | **26.71** | **25.42** | **62.62** | **36.06** | **33.33** | **31.24** |
| | | TRADES-AWP | **56.66** | 29.47 | 25.93 | 25.05 | 63.71 | 35.88 | 32.25 | 31.59 |
| | | +Ours | 55.81 | **30.27** | **26.46** | **25.54** | **64.23** | **36.77** | **32.95** | **31.99** |
| | WRN | Vanilla AT | 59.13 | 32.88 | 30.97 | 28.67 | **71.26** | 35.31 | 36.31 | 33.76 |
| | | +Ours | **60.20** | **33.24** | **31.44** | **29.15** | 71.08 | **36.62** | **37.47** | **34.96** |
| | | TRADES | 59.52 | 31.09 | 29.18 | 27.94 | **67.52** | 38.56 | 36.34 | 35.36 |
| | | +Ours | **61.25** | **33.22** | **29.65** | **28.43** | 67.51 | **39.22** | **36.91** | **35.79** |
| | | MART | **57.46** | 33.28 | 29.57 | 28.09 | **68.84** | 38.80 | 37.07 | 34.85 |
| | | +Ours | 57.07 | **33.41** | **30.28** | **28.69** | 68.75 | **39.28** | **37.93** | **35.54** |
| | | TRADES-AWP | **60.69** | 32.40 | 29.42 | 28.51 | 67.25 | 39.03 | 36.18 | 35.39 |
| | | +Ours | 59.83 | **32.72** | **29.93** | **28.98** | **67.93** | **39.59** | **36.79** | **35.64** |

AT demonstrates prominence in terms of robust accuracy comparing with corresponding baseline methods. This indicates that our approach can further boost robustness with the help of synthetic data. Notably, the averaged improvement of robust accuracy against AutoAttack with synthetic data is 0.71%, which is higher than that of models trained without synthetic data (0.52%). This may due to that additional data help the models estimate the centroid of each adversarial cluster more precisely, enforcing $\hat{I}(\epsilon)$ to be constrained more effectively and leading to better improvements in robust performance.

## 5.3 Results on Attack with Various Perturbation Budgets

As a by-product, we found that models trained with our method exhibit enhanced robust performance across different perturbation budget levels in comparison with baseline methods with a maximum accuracy gain of 3.48% (detail shown in Appendix F). This indicates employing our method can generally improve robustness independent of perturbation budget.

## 6 Conclusion

In this paper, we have introduced three new information quantities—robust CMI, separation and NCMI—to gauge the robust performance of DNNs. Based on these robust performance metrics, we have developed a new generic adversarial training framework and alternating learning algorithms to jointly minimize robust NCMI and the conventional adversarial training objective functions. Extensive experimental results show that our method consistently improves model robustness against various white-box attacks when combined with existing adversarial loss functions, demonstrating the 'plug-and-play' nature and effectiveness of our method in the field of AT.

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

## A  PROOF OF THEOREM 1

With reference to Fig. 1, it follows that $Y \to X' \to \hat{Y}'$ forms a Markov chain in the indicated order. Replacing $(X, \hat{Y})$ in Theorem 1 of Yang et al. (2023) by $(X', \hat{Y}')$, we have

$$
\begin{aligned}
\xi' = \Pr(\hat{Y}' \neq Y) &\leq \mathbb{E}_{X'} \left[ H \left( P_{Y|X'}(\cdot|X'), f(X') \right) \right] \\
&= \mathbb{E}_{X'} \left[ -\sum_{i=1}^{C} P_{Y|X'}(i|X') \cdot \ln f(X')[i] \right] \\
&= \mathbb{E}_{X'Y} \left[ -\ln f(X')[Y] \right] \\
&= \mathbb{E}_{XY} \left[ -\ln f(G(X,Y))[Y] \right].
\end{aligned}
\tag{42}
$$

Thus,

$$
\begin{aligned}
\xi_r(\epsilon) &= \max_{G: \|x' - x\|_p \leq \epsilon} \Pr(\hat{Y}' \neq Y) \\
&\leq \max_{G: \|x' - x\|_p \leq \epsilon} \mathbb{E}_{XY} \left[ -\ln f(G(X,Y))[Y] \right] \\
&= \max_{G: \|x' - x\|_p \leq \epsilon} \sum_{x,y} P(x,y) \left( -\ln f(G(x,y))[y] \right) \\
&= \sum_{x,y} P(x,y) \max_{G: \|x' - x\|_p \leq \epsilon} \left( -\ln f(G(x,y))[y] \right) \\
&= \sum_{x,y} P(x,y) \left[ \max_{x': \|x' - x\|_p \leq \epsilon} -\ln f(x')[y] \right].
\end{aligned}
\tag{43}
$$

This completes the proof of Theorem 1.

## B  PROOF OF THEOREM 2

For simplicity, we drop the subscript $\theta$ from $f_\theta(\cdot)$, and assume that the space of $X$ is discrete. With reference to Figure 1, note that

$$
P_{\hat{Y}'|XY}(\hat{Y}' = i|x, y) = f(G(x,y))[i].
\tag{44}
$$

In view of equation 44, we introduce a dummy conditional "backward channel" $\{q(x|i,y)\}$ and derive a new expression for $I(X; \hat{Y}'|Y)$ as follows:

$$
I(X; \hat{Y}'|Y) = \sum_{y} P(y) \sum_{x} P_{X|Y}(x|y) \sum_{i=1}^{C} f(G(x,y))[i] \ln \frac{f(G(x,y))[i]}{P_{\hat{Y}'|Y}(i|y)}
\tag{45}
$$

$$
= \sum_{y} P(y) \sum_{i=1}^{C} P_{\hat{Y}'|Y}(i|y) \sum_{x} \frac{P_{X|Y}(x|y) \cdot f(G(x,y))[i]}{P_{\hat{Y}'|Y}(i|y)}
$$
$$
\cdot \ln \frac{P_{X|Y}(x|y) \cdot f(G(x,y))[i]}{P_{X|Y}(x|y) P_{\hat{Y}'|Y}(i|y)}
$$

$$
= \sum_{y} P(y) \sum_{i=1}^{C} P_{\hat{Y}'|Y}(i|y) \max_{\{q(x|i,y)\}} \sum_{x} \frac{P_{X|Y}(x|y) \cdot f(G(x,y))[i]}{P_{\hat{Y}'|Y}(i|y)} \cdot \ln \frac{q(x|i,y)}{P_{X|Y}(x|y)}
\tag{46}
$$

$$
= \max_{\{q(x|i,y)\}} \sum_{x,y} P(x,y) \sum_{i=1}^{C} f(G(x,y))[i] \cdot \ln \frac{q(x|i,y)}{P_{X|Y}(x|y)}
\tag{47}
$$

where equation 45 is due to equation 44, equation 46 follows from the cross entropy inequality, and the maximization in equation 46 and equation 47 is achieved when

$$
q(x|i,y) = \frac{P_{X|Y}(x|y) \cdot f(G(x,y))[i]}{P_{\hat{Y}'|Y}(i|y)}.
\tag{48}
$$

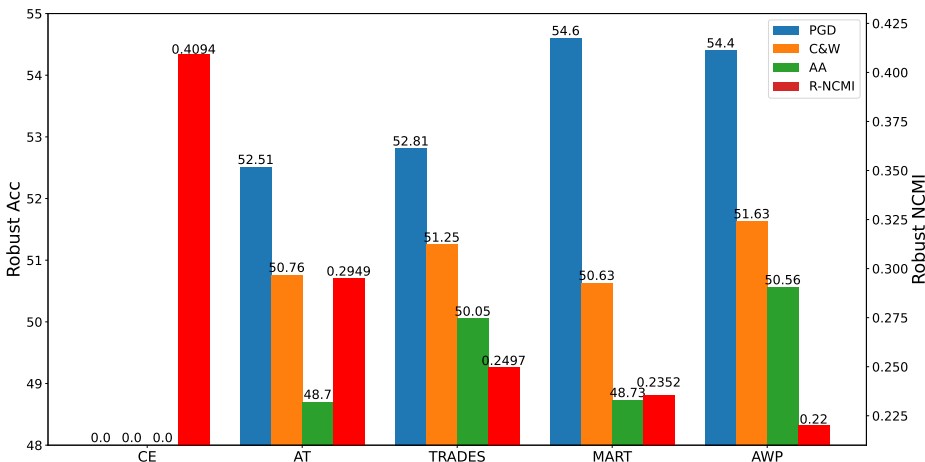

Figure 3: Robust accuracy of different learned models against PGD, C&W and AutoAttack vs the corresponding robust NCMI on CIFAR-10, where robust accuracy against PGD, C&W, and Au-toAttack are presented as blue, orange and green bars (left axis), respectively. Red bars represent corresponding robust NCMI value (right axis) of each learned model. Specific value of each evaluation metric is shown on top of each bar.

Now plugging equation 47 into equation 25, we have:

$$
I(\epsilon) = \max_{G:\|x'-x\|_p \leq \epsilon} \max_{\{q(x|i,y)\}} \sum_{x,y} P(x,y) \Big[ \sum_{i=1}^{C} f(G(x,y))[i] \cdot \ln q(x|i,y) - \ln P_{X|Y}(x|y) \Big]
$$

$$
= \max_{\{q(x|i,y)\}} \sum_{x,y} P(x,y) \left[ \max_{G:\|x'-x\|_p \leq \epsilon} \sum_{i=1}^{C} f(G(x,y))[i] \cdot \ln q(x|i,y) - \ln P_{X|Y}(x|y) \right]
$$

$$
= \max_{\{q(x|i,y)\}} \sum_{x,y} P(x,y) \left[ \max_{x':\|x'-x\|_p \leq \epsilon} \sum_{i=1}^{C} f(x')[i] \cdot \ln q(x|i,y) - \ln P_{X|Y}(x|y) \right] \quad (49)
$$

This completes the proof of equation 29. Plugging equation 47 into equation 27, we have:

$$
\hat{I}(\epsilon) = \max_{G:\|x'-x\|_p \leq \epsilon} \frac{\max\limits_{\{q(x|i,y)\}} \sum\limits_{x,y} P(x,y) \left[ \sum_{i=1}^{C} f_\theta(G(x,y))[i] \ln q(x|i,y) - \ln P_{X|Y}(x|y) \right]}{\mathbb{E}\left[ I_{\{Y \neq V\}} H(f_\theta(G(X,Y)), f_\theta(G(U,V))) \right]}
$$

$$
= \max_{\{q(x|i,y)\}} \max_{G:\|x'-x\|_p \leq \epsilon} \frac{\sum\limits_{x,y} P(x,y) \left[ \sum_{i=1}^{C} f_\theta(G(x,y))[i] \ln q(x|i,y) - \ln P_{X|Y}(x|y) \right]}{\mathbb{E}\left[ I_{\{Y \neq V\}} H(f_\theta(G(X,Y)), f_\theta(G(U,V))) \right]} \quad (50)
$$

where equation 50 is due to the fact that the denominator therein does not depend on $\{q(x|i,y)\}$. This completes the proof of equation 30 and hence Theorem 2.

In order to evaluate the robust NCMI $\hat{I}(\epsilon)$ of a given trained DNN, one should follow equation 50 to compute the corresponding value over validation data. Since we formulate equation 50 as a double maximization, we can find the optimum of $\hat{I}(\epsilon)$ through an alternating algorithm. In this alternating algorithm, we do the following steps for 5 iterations until we see convergence on the value of $\hat{I}(\epsilon)$: (1) we firstly fix $\{q(x|i,y)\}$ and solve the optimization of attack function $G$ over all validation data simultaneously due to the inter-dependency among samples when computing the denominator in equation 50; (2) we fix the attack function $G$ to compute each $q(x|i,y)$ following equation 48.

We then reported the corresponding $\hat{I}(\epsilon)$ values of different conventional AT methods in Fig. 3 in comparison with the robust accuracy against multiple white-box attacks. Indeed, the relationship between robust accuracy and robust NCMI $\hat{I}(\epsilon)$ are generally the same as that between robust accuracy and adversarial NCMI shown in Fig. 2, where a high robust accuracy matches to a low $\hat{I}(\epsilon)$. This indicates that robust NCMI $\hat{I}(\epsilon)$ can also be considered as an evaluation metric that reflects the robustness other than the intrinsic mapping structure of a DNN.

## C   PROOF OF THEOREM 3

Go back to equation 36. We introduce a dummy adversarial centroid distribution $Q'_y$ for each label $y$ and derive a new expression for $\mathbb{E}[D(f_\theta(G(X,Y))\|s'_Y)]$ as follows:

$$
\mathbb{E}[D(f_\theta(G(X,Y))\|s'_Y)] = \sum_y P(y) \sum_x P_{X|Y}(x|y) D(f_\theta(G(x,y))\|s'_y)
$$

$$
= \sum_y P(y) \Big[ \Big( \sum_x P_{X|Y}(x|y) D(f_\theta(G(x,y))\|Q'_y) \Big) - D(s'_y\|Q'_y) \Big] \tag{51}
$$

$$
= \sum_y P(y) \min_{Q'_y} \Big\{ \sum_x P_{X|Y}(x|y) D(f_\theta(G(x,y))\|Q'_y) \Big\} \tag{52}
$$

$$
= \min_{\{Q'_c\}_{c\in[C]}} \sum_y P(y) \sum_x P_{X|Y}(x|y) D(f_\theta(G(x,y))\|Q'_y)
$$

$$
= \min_{\{Q'_c\}_{c\in[C]}} \mathbb{E}[D(f_\theta(G(X,Y))\|Q'_Y)] \tag{53}
$$

where equation 52 is due to the nonnegativity of the KL divergence $D(s'_y\|Q'_y)$, and the minimization in the above is achieved when

$$
Q'_y = s'_y, \forall y \in [C]. \tag{54}
$$

Now plugging equation 53 into equation 36 yields

$$
\min_\theta \Big\{ \mathbb{E}[\max_{G_1:\|\hat{x}-x\|_p\leq\epsilon} \mathcal{L}(X,\hat{X},Y)] + \max_{G:\|x'-x\|_p\leq\epsilon} \Big[ \alpha \cdot I(X';\hat{Y}'|Y) - \beta \cdot \Gamma' \Big] \Big\}
$$

$$
= \min_\theta \max_{G_1} \max_G \Big\{ \mathbb{E}[\mathcal{L}(X,G_1(X,Y),Y)] + \alpha \cdot \min_{\{Q'_c\}_{c\in[C]}} \mathbb{E}[D(f_\theta(G(X,Y))\|Q'_Y)]
$$

$$
- \beta \cdot \mathbb{E}\left[ I_{\{Y\neq V\}} H(f_\theta(G(X,Y)), f_\theta(G(U,V))) \right] \Big\}
$$

$$
= \min_\theta \max_{G_1,G} \min_{\{Q'_c\}_{c\in[C]}} \Big\{ \mathbb{E}[\mathcal{L}(X,G_1(X,Y),Y)] + \alpha \cdot \mathbb{E}[D(f_\theta(G(X,Y))\|Q'_Y)]
$$

$$
- \beta \cdot \mathbb{E}\left[ I_{\{Y\neq V\}} H(f_\theta(G(X,Y)), f_\theta(G(U,V))) \right] \Big\}
$$

$$
\leq \min_\theta \min_{\{Q'_c\}_{c\in[C]}} \max_{G_1,G} \{ \mathbb{E}[\mathcal{L}(X,G_1(X,Y),Y)] + \alpha \cdot \mathbb{E}[D(f_\theta(G(X,Y))\|Q'_Y)]
$$

$$
- \beta \cdot \mathbb{E}\left[ I_{\{Y\neq V\}} H(f_\theta(G(X,Y)), f_\theta(G(U,V))) \right] \}. \tag{55}
$$

This completes the proof of Theorem 3.

# D   ALTERNATING LEARNING ALGORITHM

---

**Algorithm 1** $\hat{I}(\epsilon)$ Constrained Adversarial Training

---

**Input**: DNN $f$ with model parameter $\theta$, hyper-parameters $\alpha$ and $\beta$, dataset $\mathcal{D}:(\mathcal{X}, \mathcal{Y})$, number of epochs $T$, number of classes $C$, adversarial loss function $\mathcal{L}$, perturbation budget $\epsilon$, EMA factor $\tau$

1: **for** $i \in [1, \ldots, C]$ **do**
2:     Initialize adversarial centroids: $Q'_i \leftarrow \frac{1}{|\mathcal{D}_i|} \sum_{j \in \mathcal{D}_i} f_\theta(x_j)$
3: **end for**
4: **for** $e \in [1, \ldots, T]$ **do**
5:     **for** each mini-batch $\mathcal{B}$ **do**
6:         [**Update DNN parameter** $\theta$]:
7:         Determine the optimized $G_1^*$ by computing a perturbed instance $\hat{x}$ for each sample instance $(x, y) \in \mathcal{B}$ via PGD attack with 10 steps.
8:         Determine the optimized $G^*$ by computing all adversarial instances $x'$ for all sample instances $(x, y) \in \mathcal{B}$ simultaneously **all at once** in the same manner as PGD attack with 10 steps, where the objective function to be maximized in the attack process is $\mathcal{L}_\mathcal{B}(\theta, \{Q'_c\}_{c \in [C]}, G)$.
9:         $x'_j \leftarrow G^*(x_j, y_j)$ for all $(x_j, y_j) \in \mathcal{B}$
10:         $loss \leftarrow \mathcal{J}_\mathcal{B}(\theta, \{Q'_c\}_{c \in [C]}, G_1^*, G^*)$
11:         Update $\theta$ w.r.t. $loss$.
12:         [**Update adversarial centroids**]:
13:         **for** $i \in [1, \ldots, C]$ **do**
14:             Use adversarial instances $x'_j$ in line 9 to craft $\mathcal{B}'_i \leftarrow \{(x'_j, y_j) | (x_j, y_j) \in \mathcal{B}, y_j = i\}$
15:             Continue if $|\mathcal{B}'_i| = 0$
16:             $Q'_i \leftarrow \tau Q'_i + (1 - \tau) \frac{1}{|\mathcal{B}'_i|} \sum_{(x'_j, y_j) \in \mathcal{B}'_i} f_\theta(x'_j)$
17:         **end for**
18:     **end for**
19: **end for**
20: **return** DNN $f_\theta$

---

# E  RELAXED ALTERNATING LEARNING ALGORITHM

---

**Algorithm 2** Relaxed Alternating Algorithm for $\hat{I}(\epsilon)$ Constrained Adversarial Training

---

**Input**: DNN $f$ with model parameter $\theta$, hyper-parameters $\alpha$ and $\beta$, dataset $\mathcal{D}$:$(\mathcal{X}, \mathcal{Y})$, number of epochs $T$, number of classes $C$, adversarial loss function $\mathcal{L}$, perturbation budget $\epsilon$

1: **for** $e \in [1, \ldots, T]$ **do**
2:     **[Update DNN parameter $\theta$]**:
3:         **for** each mini-batch $\mathcal{B} \subset$ train set $\mathcal{D}_{\text{train}}$ **do**
4:             Determine the optimized $G_1^*$ by computing a perturbed instance $\hat{x}$ for each sample instance $(x, y) \in \mathcal{B}$ via PGD attack with 10 steps.
5:             **if** $e > 1$ **then**
6:                 Determine the optimized $G^*$ by computing all adversarial instances $x'$ for all sample instances $(x, y) \in \mathcal{B}$ simultaneously **all at once** in the same manner as PGD attack with 10 steps, where the objective function to be maximized in the attack process is $\mathcal{L}_{\mathcal{B}}(\theta, \{Q'_c\}_{c \in [C]}, G)$ in equation 40.
7:                 $x'_j \leftarrow G^*(x_j, y_j)$ for all $(x_j, y_j) \in \mathcal{B}$
8:                 $loss \leftarrow \mathcal{J}_{\mathcal{B}}(\theta, \{Q'_c\}_{c \in [C]}, G_1^*, G^*)$
9:             **else**
10:                 $loss \leftarrow \frac{1}{|\mathcal{B}|} \sum_{(x_j, y_j) \in \mathcal{B}} \mathcal{L}(x_j, G_1^*(x_j, y_j), y_j)$
11:             **end if**
12:             Update $\theta$ w.r.t. $loss$
13:         **end for**
14:     **if** $e = 1$ **then**
15:         Initialize adversarial centroids: $Q'_i \leftarrow \frac{1}{|\mathcal{D}_i|} \sum_{j \in \mathcal{D}_i} f_\theta(x_j)$ for $i \in [1, \ldots, C]$
16:     **end if**
17:     **[Update adversarial centroids]**:
18:     Initialize $Q'_{i,dummy}$ to zero vectors for $i \in [1, \ldots, C]$
19:     **for** each mini-batch $\mathcal{B}_Q \subset$ train set $\mathcal{D}_{\text{train}}$ with $|\mathcal{B}_Q| = 4 \cdot |\mathcal{B}|$ **do**
20:         Determine the optimized $G^*$ by computing all adversarial instances $x'_Q$ for all sample instances $(x_Q, y_Q) \in \mathcal{B}_Q$ simultaneously **all at once** in the same manner as PGD attack with 10 steps, where the objective function to be maximized in the attack process is $\mathcal{L}_{\mathcal{B}_Q}(\theta, \{Q'_c\}_{c \in [C]}, G)$ in equation 40.
21:         $x'_{j,Q} \leftarrow G^*(x_{j,Q}, y_{j,Q})$ for all $(x_{j,Q}, y_{j,Q}) \in \mathcal{B}_Q$
22:         Construct the set $\mathcal{B}'_Q$ consisting of all pairs $(x'_{j,Q}, y_{j,Q})$ generated in line 21
23:         **for** $(x'_{j,Q}, y_{j,Q}) \in \mathcal{B}'_Q$ **do**
24:             $Q'_{y_{j,Q},dummy} \leftarrow Q'_{y_{j,Q},dummy} + f_\theta(x'_{j,Q})$
25:         **end for**
26:     **end for**
27:     Normalize each $Q'_{i,dummy}$ to a probability distribution for $i \in [1, \ldots, C]$
28:     $Q'_i \leftarrow Q'_{i,dummy}$ for $i \in [1, \ldots, C]$
29: **end for**
30: **return** DNN $f_\theta$

---

In our relaxed alternating learning algorithm, we update the dummy centroid distributions $Q'_y$ of all adversarial clusters once per epoch rather than once per mini-batch. The pseudo-code of this algorithm is shown in Algorithm 2. In the case where there are not enough adversarial instances in each mini-batch to estimate the centroid of each adversarial cluster, this delivers a good compromise between time complexity and robust accuracy. Our experiments on the original CIFAR-100 and Tiny-ImageNet datasets were conducted using Algorithm 2 with the respective robust accuracy results reported in Table 2 and 1. In comparison with vanilla AT, TRADES, and MART, Algorithm 1 and Algorithm 2 require roughly 60% and 120% more training time in our setup, whereas TRADES-AWP requires roughly 30% more training time.

Note that in the above algorithm, when we iterate through the training set one more time to update all adversarial centroids at the end of each epoch, we increase size of each mini-batch $|\mathcal{B}_Q|$ to 4 times as that when we update DNN parameter $\theta$. This adjustment is mostly in consideration of providing

a better estimation on the average cross entropy between sample pairs with different class labels (second term in equation 40), as compute this value over a larger mini-batch can approximate this value more precisely. Ideally, one should compute this value over the entire training set in one pass. Due to the limit of computational resource, when we update adversarial centroids, we only increase the mini-batch size up to 4 times (batch size of 512) as that in conventional training (batch size of 128).

# F Robustness Against Various Perturbation Budgets

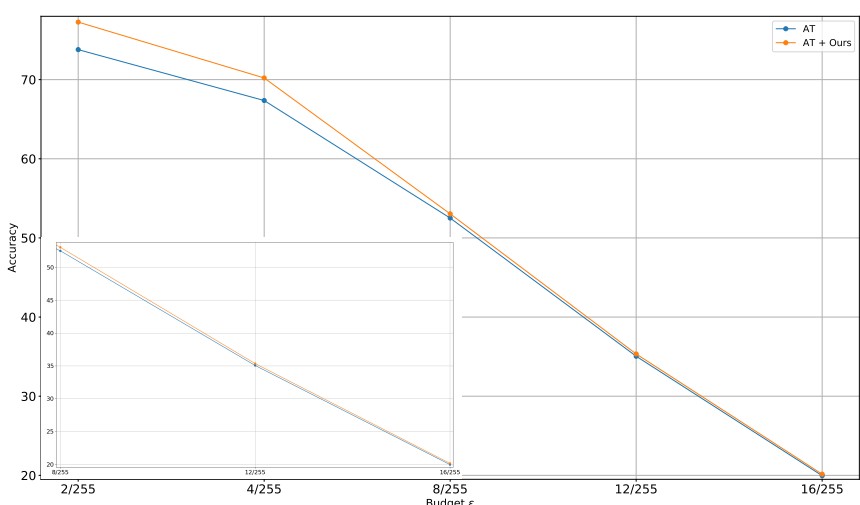

Figure 4: Robust accuracy against PGD attack with various perturbation budgets $\epsilon$ on CIFAR-10 when the underlying DNN model is PreAct-ResNet-18 and baseline method is vanilla AT.
.

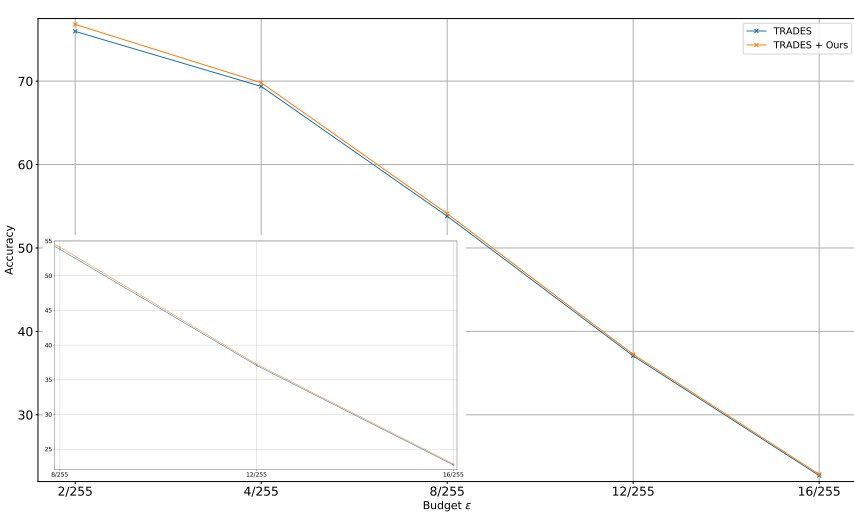

Figure 5: Robust accuracy against PGD attack with various perturbation budgets $\epsilon$ on CIFAR-10 when the underlying DNN model is PreAct-ResNet-18 and baseline method is TRADES.
.

Other than showing that our approach outperformed the selected baseline methods on robust accuracy with a fixed perturbation budget $\epsilon = 8/255$, we additionally evaluated our trained DNN against PGD attack (Madry et al., 2018) with various perturbation budgets $\epsilon$, and compared with the DNN

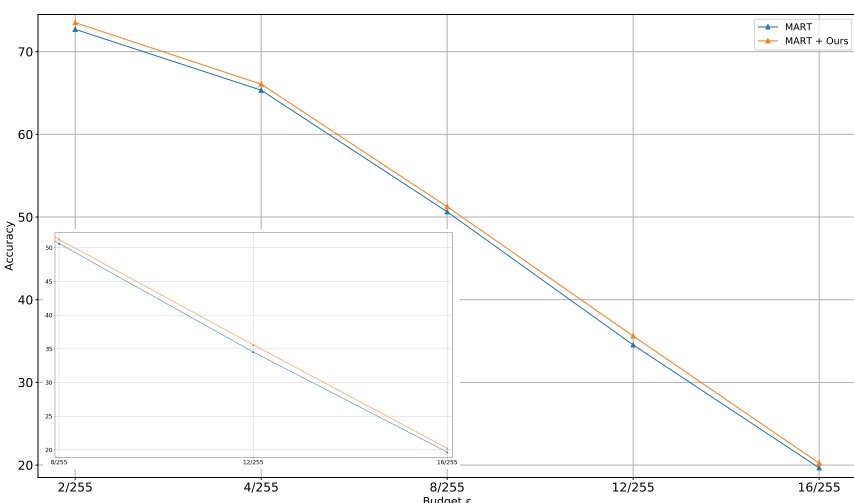

Figure 6: Robust accuracy against C&W$_\infty$ attack with various perturbation budgets $\epsilon$ on CIFAR-10 when the underlying DNN model is PreAct-ResNet-18 and baseline method is MART.

.

trained by baseline method. In Fig. 4, we chose multiple perturbation budgets, either greater or less than the standard one, to attack the DNN trained by vanilla AT (trained by TRADES and MART in Fig. 5 and 6, respectively) with and without constraining on $\hat{I}(\epsilon)$ when the underlying model architecture is PreAct-ResNet-18 and the dataset is CIFAR-10. We found that the model trained with our method consistently exhibits better robustness against different perturbation budgets comparing with the baseline counterparts. For vanilla AT, the maximum robust accuracy gain is 3.48% when $\epsilon = 2/255$, and the minimum gain is 0.21% when $\epsilon = 16/255$. When the baseline method is TRADES, the maximum robust accuracy gain is 0.84% when $\epsilon = 2/255$, and minimum gain of 0.15% when $\epsilon = 16/255$. In the case of MART, the maximum robust accuracy gain is 1.08% when $\epsilon = 12/255$, and the minimum is 0.57% when $\epsilon = 16/255$. The results in this section indicate our method are generally more robust than standard AT methods under various perturbation budgets.

## G  VISUALIZATION OF $\hat{I}(\epsilon)$ CONSTRAINED AT AND HYPER-PARAMETER SETTINGS

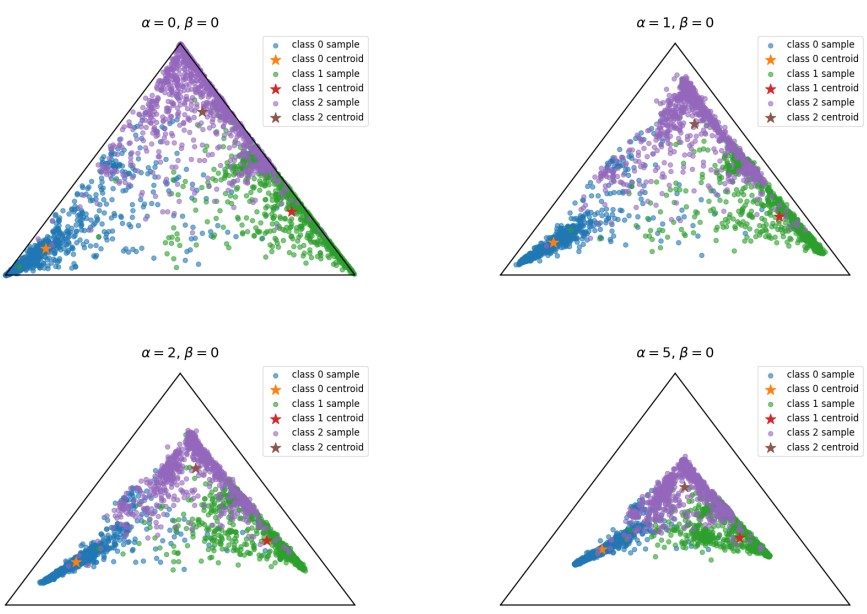

Figure 7: Visualization of the output probability simplex on the CIFAR-10 validation set, with varying constraining levels on adversarial CMI ($\alpha$) and a fixed constraining level on adversarial separation ($\beta = 0$).

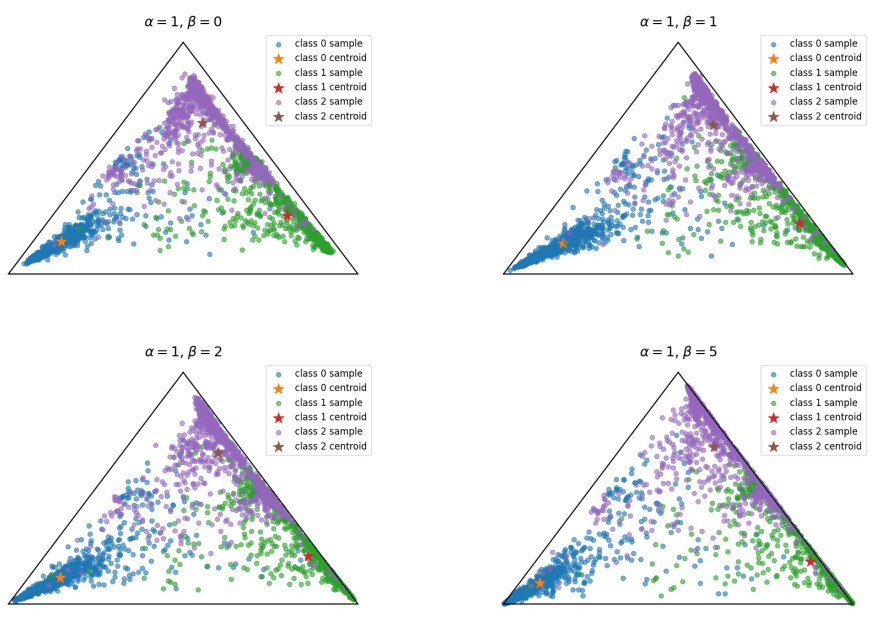

Figure 8: Visualization of the output probability simplex on the CIFAR-10 validation set, with varying constraining levels on adversarial separation ($\beta$) and a fixed constraining level on adversarial CMI ($\alpha = 1$).

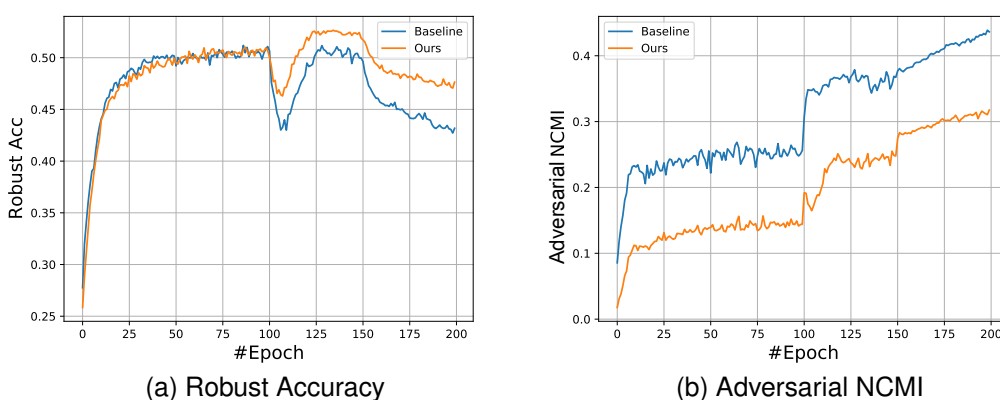

(a) Robust Accuracy        (b) Adversarial NCMI

Figure 9: Evaluation metric curves of PreAct-ResNet-18 on CIFAR-10 validation set during training. All metrics are computed over validation data after C&W$_\infty$ attack with 40 steps. 'Baseline' stands for training with vanilla AT (Madry et al., 2018), and 'Ours' represent vanilla AT with robust NCMI regularization.

### G.1 SIMPLEX VISUALIZATION

In this subsection, we visualize the effect of our adversarial CMI and adversarial separation under different constraining levels when the underlying DNN architecture is PreAct-ResNet-18 (He et al., 2016b). We first randomly picked 3 classes from the CIFAR-10 validation set, then focused on a subset of validation samples with these labels perturbed by C&W$_\infty$ attack (Carlini & Wagner, 2017) in 40 steps. Each sample from this subset is fed into the DNN, and only the three logits corresponding to the selected 3 labels are retained. These logits are subsequently converted into a 3-dimensional probability vector using the softmax operation. After the steps mentioned above, we further projected all obtained 3-dimensional probability vectors from the samples in the constructed validation subset into a 2-dimensional simplex to visualize the concentration and separation effect of our trained robust DNNs in Fig. 7 and 8.

We compare the effect of varing constraining levels on adversarial CMI and separation, utilizing vanilla AT (Madry et al., 2018) as the base adversarial loss function, in Fig. 7 and 8, respectively. With different constraining levels on adversarial CMI, one can observe that a larger value of $\alpha$ enforce each adversarial cluster more compact around its centroid, while different clusters are closer to each other at the same time. We suggest that this creates a trade-off, as a stronger constraining level on intra-class concentration (adversarial CMI) might weaken the regularization effect on robust accuracy. This also justifies the necessity of constraining on adversarial separation, which further separates perturbed samples with different labels apart. As shown in Fig. 8, a stronger constraining level on adversarial separation (larger $\beta$) enlarge the margins between each pair of adversarial clusters, and further push them to the corresponding corner point with a negligible affect on concentration within each adversarial cluster.

In general, since our method enforces data points from the same class to be more concentrated around the centroid of corresponding cluster and different clusters to be more separated from each other, the overlapping areas in the simplex among different classes presented above are smaller than that of the baseline method (top-left subplots of Fig. 7 with title '$\alpha = 0$, $\beta = 0$'), especially when $\alpha = 1$ and $\beta$ is set to 2 or 5 (bottom row of Fig. 8). As such, there are less data points near the decision boundaries and susceptible against adversarial attacks under our proposed framework comparing with baseline, thereby enhances the robustness of a model.

### G.2 TRAINING CURVES OF $\hat{I}(\epsilon)$ CONSTRAINED AT

In Fig. 9, we present the evolution curves of robust accuracy and adversarial NCMI of PreAct-ResNet-18 on CIFAR-10 validation data perturbed by C&W$_\infty$ attack in 40 steps, using loss function

of vanilla AT with and without constraining on $\hat{I}(\epsilon)$. In Fig. 9a, robust accuracy of vanilla AT with and without our regularization is almost the same before the first learning rate decay at the $100^{\text{th}}$ epoch. After that, the robust accuracy of our method is always better than that of baseline. Moreover, the gap between the two curves increases towards the end of the training phase, indicating that our approach also suffers less from robust overfitting. The curves for adversarial NCMI are displayed in Fig. 9b, where the adversarial NCMI of our approach shows a significant gap from the first epoch and remains consistently lower than the baseline. This demonstrates that our method exhibits better intra-class concentration and inter-class separation for perturbed samples. One may notice that in Fig. 9b, both curves show an upward trend near the end of training phase. We argue that this is due to robust overfitting, as robust accuracy on validation set of both curves continue to drop after the last learning rate decay at the $150^{\text{th}}$ epoch.

G.3   HYPER-PARAMETER SETTINGS

Table 3: Hyper-parameter settings of $\hat{I}(\epsilon)$ constrained AT on CIFAR-{10, 100} datasets. Each parenthesis represent the hyper-parameter value of $\alpha$ and $\beta$ in our approach, respectively.

| Dataset | Method | No Synthetic Data | | 1M Synthetic Data | |
|---|---|---|---|---|---|
| | | **PreAct-RN18** | **WRN-34-10** | **PreAct-RN18** | **WRN-28-10** |
| **CIFAR-10** | Vanilla AT | (1, 2) | (1, 2) | (2, 0.1) | (2, 1) |
| | TRADES | (0.01, 0.1) | (0.01, 0.1) | (0.1, 2) | (0.1, 2) |
| | MART | (0.01, 2) | (0.1, 1) | (0.1, 1) | (0.1, 1) |
| | TRADES-AWP | (0.1, 1) | (0.1, 1) | (0.1, 2) | (0.1, 1) |
| **CIFAR-100** | Vanilla AT | (2, 5) | (2, 5) | (0.1, 2) | (0.1, 2) |
| | TRADES | (1, 2) | (1, 2) | (0.01, 5) | (0.01, 2) |
| | MART | (0.1, 2) | (0.1, 0.01) | (0.1, 5) | (0.1, 2) |
| | TRADES-AWP | (0.01, 5) | (0.01, 5) | (0.1, 2) | (0.1, 1) |

Table 4: Hyper-parameter settings of $\hat{I}(\epsilon)$ constrained AT on Tiny-ImageNet datasets. Each parenthesis represent the hyper-parameter value of $\alpha$ and $\beta$ in our approach, respectively.

| Method | Model | |
|---|---|---|
| | **PreAct-RN18** | **PreAct-RN34** |
| Vanilla AT | (1, 0.1) | (1, 0.1) |
| TRADES | (0.01, 2) | (0.01, 2) |

In this subsection, we reported the hyper-parameter settings of our approach, which is the values of $\alpha$ and $\beta$ in our objective function in equation 39. Table 3 and 4 provide the specific values of $\alpha$ and $\beta$ used in our experiments on the CIFAR and Tiny-ImageNet datasets, respectively. As one can see, for the same AT method on same dataset, the hyper-parameter settings of our approach are generally the same between different models, while those between methods and datasets are somehow different. We attribute this to the variation in underlying data distributions across datasets and the differences in the DNN's intrinsic mapping structures between methods, which result in different optimal hyper-parameter settings for the same model architecture on each method and dataset.

# H    Ablation Study

In this section, we conducted ablation study on original CIFAR-10 dataset to verify the effectiveness of each individual term in equation 40. We selected PreAct-ResNet-18 (He et al., 2016b) as the underlying DNN model and vanilla AT (Madry et al., 2018) as the baseline method. We present our experimental results in Table 5.

Table 5: Experimental results of ablation study on original CIFAR-10 dataset. Best results are highlighted in **bold**.

| Hyper-Param | Clean | PGD | C&W$_\infty$ | AA |
|---|---|---|---|---|
| $\alpha = 0, \beta = 0$ (Baseline) | 79.35 | 52.51 | 50.76 | 48.70 |
| $\alpha = 0.001, \beta = 0$ | 82.25 | 52.56 | 50.93 | 48.73 |
| $\alpha = 0.01, \beta = 0$ | 81.95 | 52.27 | 50.86 | 48.79 |
| $\alpha = 0.1, \beta = 0$ | **82.63** | 52.65 | 50.79 | 48.82 |
| $\alpha = 1, \beta = 0$ | 81.32 | **52.83** | **51.99** | **49.39** |
| $\alpha = 2, \beta = 0$ | 80.37 | 52.61 | 51.51 | 49.01 |
| $\alpha = 5, \beta = 0$ | 79.96 | 52.78 | 50.94 | 48.99 |
| $\alpha = 0, \beta = 0.001$ | 79.49 | 52.60 | 50.79 | 48.72 |
| $\alpha = 0, \beta = 0.01$ | 79.43 | **52.79** | 50.80 | 48.74 |
| $\alpha = 0, \beta = 0.1$ | **81.72** | 52.54 | 51.38 | 48.79 |
| $\alpha = 0, \beta = 1$ | 80.93 | 52.76 | 52.12 | 49.34 |
| $\alpha = 0, \beta = 2$ | 80.72 | 52.59 | **52.18** | **49.35** |
| $\alpha = 0, \beta = 5$ | 79.72 | 52.61 | 51.94 | 49.30 |
| $\alpha = 1, \beta = 2$ (Best) | 81.43 | **53.04** | **52.23** | **49.47** |

As shown in the table above, we found that solely enabling each term in equation 40 can enhance the robustness of a DNN model in comparison with the baseline method. In addition, when enabling both terms by selecting the corresponding hyper-parameter values of $\alpha$ and $\beta$ with the highest accuracy against AutoAttack (Croce & Hein, 2020) can further improve the robust performance. This demonstrates the efficacy of both proposed metrics in our work.

# I    Additional Experimental Results

## I.1    Results Against $L_2$-norm Attack

Table 6: Experimental results of our trained models against $L_2$-norm attacks on original CIFAR-10 dataset.

| Method | PreAct-RN18 | | WRN-34-10 | |
|---|---|---|---|---|
| | PGD-$L_2$ | C&W-$L_2$ | PGD-$L_2$ | C&W-$L_2$ |
| Vanilla AT | 62.22 | 60.19 | 63.61 | 63.07 |
| +Ours | 63.85 | 62.13 | 65.27 | 63.23 |
| TRADES | 63.03 | 61.47 | 63.48 | 62.45 |
| +Ours | 63.53 | 61.97 | 63.93 | 62.79 |
| MART | 63.75 | 60.29 | 64.93 | 62.49 |
| +Ours | 63.60 | 61.06 | 64.73 | 63.09 |
| TRADES-AWP | 63.23 | 61.19 | 64.56 | 62.92 |
| +Ours | 63.76 | 61.45 | 64.81 | 63.51 |

In this subsection, we evaluate our DNNs, which **trained with adversarially perturbed data in $L_\infty$-norm** on original CIFAR-10 dataset, against data attacked in $L_2$-norm. We selected PGD-$L_2$ (Madry et al., 2018) and C&W-$L_2$ (Carlini & Wagner, 2017) as our evaluation attack methods, where both attacks are crafted in 20 iterations and have a budget $\epsilon = 128/255$ with a step size of $15/255$. We present our experimental results in Table 6, where the robust accuracy of our approach outperforms corresponding baseline methods in most cases. The results indicate that the models trained with our method exhibit superiority against $L_2$-norm attack even though such models are not

explicitly trained with data perturbed in $L_2$-norm, and we believe the improvement against $L_2$-norm attacks could be even more substantial if the models are trained with data perturbed in $L_2$-norm.

## I.2 COMPARISON OF ROBUST PERFORMANCE BETWEEN BEST AND LAST CHECKPOINTS

Table 7: Experimental results of best and last checkpoints trained with baseline methods and our approach on original CIFAR-10 dataset. Better results are highlighted in **bold**.

| Model | Method | Best | | | | Last | | | |
|-------|--------|-------|------|------------|------|-------|------|------------|------|
| | | Clean | PGD | C&W$_\infty$ | AA | Clean | PGD | C&W$_\infty$ | AA |
| PreAct-RN18 | Vanilla AT | 79.35 | 52.51 | 50.76 | 48.70 | 78.89 | 43.18 | 44.08 | 41.53 |
| | +Ours | **81.43** | **53.04** | **52.23** | **49.47** | **80.66** | **46.63** | **47.32** | **43.98** |
| | TRADES | 82.21 | 53.81 | 51.25 | 50.05 | 81.57 | 50.47 | 49.26 | 47.40 |
| | +Ours | **82.26** | **54.12** | **51.65** | **50.70** | **82.20** | **50.77** | **49.67** | **48.19** |
| | MART | **79.81** | **54.60** | 50.63 | 48.73 | **79.58** | 47.26 | 45.78 | 42.71 |
| | +Ours | 79.40 | 54.41 | **51.25** | **49.61** | 79.22 | **47.52** | **46.04** | **43.66** |
| | TRADES-AWP | 81.52 | 54.40 | 51.63 | 50.56 | 81.22 | 53.14 | 50.97 | 49.71 |
| | +Ours | **81.73** | **54.64** | **52.15** | **51.07** | **81.40** | **53.71** | **51.56** | **50.64** |
| WRN-34-10 | Vanilla AT | 85.06 | 56.23 | 56.00 | 53.58 | 84.05 | 46.93 | 47.73 | 45.38 |
| | +Ours | **85.58** | **57.25** | **56.20** | **54.22** | **84.48** | **47.16** | **48.20** | **46.15** |
| | TRADES | **85.54** | 57.19 | 56.26 | 55.03 | **85.44** | 48.98 | 50.06 | 47.82 |
| | +Ours | 85.24 | **57.56** | **56.60** | **55.27** | 85.08 | **49.14** | **50.21** | **48.22** |
| | MART | **84.15** | **57.64** | 55.40 | 53.55 | **83.63** | **48.06** | 47.71 | 45.04 |
| | +Ours | 83.99 | 57.20 | **55.92** | **53.77** | 83.58 | 48.05 | **47.90** | **45.46** |
| | TRADES-AWP | **85.14** | 58.02 | 56.66 | 55.35 | **84.17** | 50.52 | 51.06 | 48.94 |
| | +Ours | 84.67 | **58.25** | **56.81** | **55.60** | 84.05 | **50.84** | **51.23** | **49.34** |

In this subsection, we report the clean and robust accuracy of the best and last checkpoints for models trained with the selected baseline methods and combining with our approach on original CIFAR-10 dataset in Table 7. We found that the robust performances of models trained with our method are better than the baseline counterparts. Furthermore, the robust accuracy gap between best and last checkpoints trained by our approach is less significant than that of corresponding baseline methods for most pairs of comparisons, which demonstrates that models trained with our method are less likely to suffer from overfitting than baseline methods.

## I.3 RESULTS AGAINST BLACK-BOX ATTACK

Table 8: Experimental results of PreAct-ResNet-18 against Square attack on original CIFAR-10 dataset.

| Method | Clean Acc. | Robust Acc. |
|--------|------------|-------------|
| Vanilla AT | 79.35 | 55.00 |
| +Ours | 83.43 | 56.87 |
| TRADES | 82.21 | 56.27 |
| +Ours | 82.26 | 57.01 |
| MART | 79.81 | 54.84 |
| +Ours | 79.40 | 55.71 |
| TRADES-AWP | 81.52 | 56.03 |
| +Ours | 81.73 | 56.71 |

In this subsection, we evaluated the performance of models trained with baseline methods and our approach against black-box attack. We selected Square attack (Andriushchenko et al., 2020) as the evaluation attack method, and present the results of PreAct-ResNet-18 (He et al., 2016b) against this attack on original CIFAR-10 dataset in Table 8, where the black-box robust accuracy of our approach consistently outperform that of corresponding baseline methods.

## I.4 COMPARISON WITH OTHER METHODS UTILIZING MUTUAL INFORMATION

In this subsection, we present our results of our method comparing with other approaches which improve adversarial robustness using mutual information like HBaR (Wang et al., 2021) and IB-RAR

(Xu et al., 2023a). In order to make a fair comparison with such methods, we run all experiments of the above approaches and ours under the experimental setting described in Section 5 on original CIFAR-10 dataset where the DNN architecture is ResNet18 (He et al., 2016a). We combine HBaR, IB-RAR and our method with vanilla AT (Madry et al., 2018) and TRADES (Zhang et al., 2019), and report the results of best checkpoints in Table 9, where our method is more robust against AutoAttack (Croce & Hein, 2020) than other approaches.

Table 9: Experimental results of methods utilizing mutual information to improve adversarial robustness on original CIFAR-10 dataset with DNN architecture of ResNet18.

| Method | Clean | PGD | C&W$_\infty$ | AA |
|---|---|---|---|---|
| Vanilla AT | 83.65 | 51.64 | 50.66 | 48.10 |
| +HBaR | **83.96** | 52.49 | 50.70 | 48.55 |
| +IB-RAR | 83.71 | 52.53 | 50.73 | 48.63 |
| +Ours | 83.04 | **52.80** | **51.06** | **48.79** |
| TRADES | 83.36 | 51.93 | 50.40 | 49.00 |
| +HBaR | 83.39 | 52.49 | **51.09** | 49.30 |
| +IB-RAR | **83.44** | 52.58 | 50.67 | 49.56 |
| +Ours | 82.75 | **52.87** | 51.05 | **50.09** |

## J Variance of Experimental Results

Table 10: Experimental results and variances on CIFAR-{10, 100} datasets, both with and without synthetic data, are averaged over 3 runs. 'Clean' refers to accuracy on benign samples, and 'AA' represents robust accuracy against AutoAttack. '+Ours' indicates the use of our $\hat{I}(\epsilon)$ constrained AT. Results for original datasets are listed under 'No Synthetic Data', while those with additional synthetic data are under '1M Synthetic Data'. WRN-34-10 was used only for original datasets (in red cells), and WRN-28-10 only for datasets with synthetic data (in blue cells). Better results are in **bold**.

| Dataset | Model | Method | No Synthetic Data | | 1M Synthetic Data | |
|---|---|---|---|---|---|---|
| | | | Clean | AA | Clean | AA |
| CIFAR-10 | PreAct-RN18 | Vanilla AT | $79.35 \pm 0.12$ | $48.70 \pm 0.17$ | $\mathbf{90.08 \pm 0.16}$ | $54.91 \pm 0.12$ |
| | | +Ours | $\mathbf{81.43 \pm 0.15}$ | $\mathbf{49.47 \pm 0.20}$ | $89.88 \pm 0.19$ | $\mathbf{56.61 \pm 0.08}$ |
| | | TRADES | $82.21 \pm 0.09$ | $50.05 \pm 0.13$ | $\mathbf{88.48 \pm 0.06}$ | $58.13 \pm 0.06$ |
| | | +Ours | $\mathbf{82.26 \pm 0.05}$ | $\mathbf{50.70 \pm 0.15}$ | $87.90 \pm 0.05$ | $\mathbf{58.52 \pm 0.08}$ |
| | | MART | $\mathbf{79.81 \pm 0.13}$ | $48.73 \pm 0.11$ | $86.99 \pm 0.05$ | $52.86 \pm 0.03$ |
| | | +Ours | $79.40 \pm 0.18$ | $\mathbf{49.61 \pm 0.10}$ | $\mathbf{87.82 \pm 0.04}$ | $\mathbf{53.92 \pm 0.03}$ |
| | | TRADES-AWP | $81.52 \pm 0.09$ | $50.56 \pm 0.06$ | $87.69 \pm 0.05$ | $57.54 \pm 0.10$ |
| | | +Ours | $\mathbf{81.73 \pm 0.06}$ | $\mathbf{51.07 \pm 0.06}$ | $\mathbf{87.73 \pm 0.09}$ | $\mathbf{58.25 \pm 0.07}$ |
| | WRN | Vanilla AT | $85.06 \pm 0.05$ | $53.58 \pm 0.10$ | $\mathbf{92.69 \pm 0.07}$ | $61.47 \pm 0.08$ |
| | | +Ours | $\mathbf{85.58 \pm 0.06}$ | $\mathbf{54.22 \pm 0.13}$ | $92.05 \pm 0.08$ | $\mathbf{61.71 \pm 0.07}$ |
| | | TRADES | $\mathbf{85.54 \pm 0.10}$ | $55.03 \pm 0.08$ | $90.26 \pm 0.03$ | $62.73 \pm 0.03$ |
| | | +Ours | $85.24 \pm 0.05$ | $\mathbf{55.27 \pm 0.05}$ | $\mathbf{90.75 \pm 0.10}$ | $\mathbf{63.27 \pm 0.07}$ |
| | | MART | $\mathbf{84.15 \pm 0.11}$ | $53.55 \pm 0.03$ | $\mathbf{91.91 \pm 0.04}$ | $61.21 \pm 0.04$ |
| | | +Ours | $83.99 \pm 0.13$ | $\mathbf{53.77 \pm 0.08}$ | $91.84 \pm 0.03$ | $\mathbf{61.75 \pm 0.05}$ |
| | | TRADES-AWP | $\mathbf{85.14 \pm 0.07}$ | $55.35 \pm 0.04$ | $\mathbf{90.44 \pm 0.06}$ | $63.17 \pm 0.08$ |
| | | +Ours | $84.67 \pm 0.09$ | $\mathbf{55.60 \pm 0.06}$ | $90.37 \pm 0.11$ | $\mathbf{63.41 \pm 0.03}$ |
| CIFAR-100 | PreAct-RN18 | Vanilla AT | $52.46 \pm 0.11$ | $24.90 \pm 0.12$ | $\mathbf{66.95 \pm 0.16}$ | $29.76 \pm 0.10$ |
| | | +Ours | $\mathbf{54.13 \pm 0.20}$ | $\mathbf{25.70 \pm 0.12}$ | $65.93 \pm 0.08$ | $\mathbf{30.88 \pm 0.11}$ |
| | | TRADES | $55.48 \pm 0.08$ | $24.36 \pm 0.13$ | $\mathbf{63.83 \pm 0.08}$ | $31.92 \pm 0.09$ |
| | | +Ours | $\mathbf{57.95 \pm 0.13}$ | $\mathbf{24.76 \pm 0.09}$ | $63.73 \pm 0.11$ | $\mathbf{32.45 \pm 0.10}$ |
| | | MART | $\mathbf{52.10 \pm 0.15}$ | $25.00 \pm 0.11$ | $62.44 \pm 0.14$ | $29.87 \pm 0.06$ |
| | | +Ours | $51.46 \pm 0.13$ | $\mathbf{25.42 \pm 0.11}$ | $\mathbf{62.62 \pm 0.04}$ | $\mathbf{31.24 \pm 0.05}$ |
| | | TRADES-AWP | $\mathbf{56.66 \pm 0.07}$ | $25.05 \pm 0.08$ | $63.71 \pm 0.05$ | $31.59 \pm 0.06$ |
| | | +Ours | $55.81 \pm 0.10$ | $\mathbf{25.54 \pm 0.05}$ | $\mathbf{64.23 \pm 0.05}$ | $\mathbf{31.99 \pm 0.09}$ |
| | WRN | Vanilla AT | $59.13 \pm 0.08$ | $28.67 \pm 0.08$ | $\mathbf{71.26 \pm 0.04}$ | $33.76 \pm 0.05$ |
| | | +Ours | $\mathbf{60.20 \pm 0.04}$ | $\mathbf{29.15 \pm 0.10}$ | $71.08 \pm 0.03$ | $\mathbf{34.96 \pm 0.11}$ |
| | | TRADES | $59.52 \pm 0.10$ | $27.94 \pm 0.09$ | $\mathbf{67.52 \pm 0.08}$ | $35.36 \pm 0.04$ |
| | | +Ours | $\mathbf{61.25 \pm 0.17}$ | $\mathbf{28.43 \pm 0.11}$ | $67.51 \pm 0.11$ | $\mathbf{35.79 \pm 0.06}$ |
| | | MART | $\mathbf{57.46 \pm 0.05}$ | $28.09 \pm 0.05$ | $\mathbf{68.84 \pm 0.12}$ | $34.85 \pm 0.05$ |
| | | +Ours | $57.07 \pm 0.06$ | $\mathbf{28.69 \pm 0.07}$ | $68.75 \pm 0.08$ | $\mathbf{35.54 \pm 0.09}$ |
| | | TRADES-AWP | $\mathbf{60.69 \pm 0.12}$ | $28.51 \pm 0.08$ | $67.25 \pm 0.09$ | $35.39 \pm 0.10$ |
| | | +Ours | $59.83 \pm 0.08$ | $\mathbf{28.98 \pm 0.07}$ | $\mathbf{67.93 \pm 0.15}$ | $\mathbf{35.64 \pm 0.04}$ |

# K  CONSTRAINING NCMI OVER BENIGN SAMPLES ALSO BOOSTS ROBUSTNESS

Table 11: Experimental results of constraining NCMI on benign samples (denoted as '+Clean NCMI') with TRADES on CIFAR-{10, 100} datasets, with and without synthetic data, in comparison with TRADES and $\hat{I}(\epsilon)$ constrained AT (denoted as '+Robust NCMI'), are averaged over 3 runs. WRN-34-10 was only used for original datasets (cells marked in red), and WRN-28-10 for datasets with synthetic data (cells marked in blue). Best result of each group are marked in **bold**.

| Dataset | Model | Method | No Synthetic Data | | | | 1M Synthetic Data | | | |
|---|---|---|---|---|---|---|---|---|---|---|
| | | | Clean | PGD | CW$_\infty$ | AA | Clean | PGD | CW$_\infty$ | AA |
| CIFAR-10 | PreAct-RN18 | TRADES | 82.21 | 53.81 | 51.25 | 50.05 | **88.48** | 61.16 | 58.87 | 58.13 |
| | | +Robust NCMI | **82.26** | 54.12 | 51.65 | 50.70 | 87.90 | **61.51** | **59.80** | **58.52** |
| | | +Clean NCMI | 81.98 | **54.94** | **52.03** | **50.99** | 88.25 | 61.37 | 59.40 | 58.42 |
| | WRN | TRADES | **85.54** | 57.19 | 56.26 | 55.03 | 90.26 | 64.95 | 63.94 | 62.73 |
| | | +Robust NCMI | 85.24 | 57.56 | **56.60** | **55.27** | 90.75 | **65.45** | **64.77** | **63.27** |
| | | +Clean NCMI | 85.39 | **57.87** | 56.45 | 55.22 | **90.78** | 65.27 | 64.48 | 63.19 |
| CIFAR-100 | PreAct-RN18 | TRADES | 55.48 | 28.55 | 25.57 | 24.36 | 63.83 | 36.08 | 32.84 | 31.92 |
| | | +Robust NCMI | **57.95** | **29.92** | 26.00 | **24.76** | 63.73 | **36.32** | **33.08** | **32.45** |
| | | +Clean NCMI | 55.93 | 29.17 | **26.02** | 24.64 | **64.69** | 36.16 | 32.85 | 32.04 |
| | WRN | TRADES | 59.52 | 31.09 | 29.18 | 27.94 | 67.52 | 38.56 | 36.34 | 35.36 |
| | | +Robust NCMI | **61.25** | **33.22** | **29.65** | **28.43** | 67.51 | 39.22 | 36.91 | 35.79 |
| | | +Clean NCMI | 59.78 | 31.50 | 29.52 | 28.32 | **67.74** | **39.23** | **37.21** | **35.99** |

In Yang et al. (2023), the authors mentioned that in benign image classification task, training a DNN with constraining NCMI on benign samples leads to improved robustness compared to training with only CE loss. Their experimental results indicate that models trained with CMIC-DL framework are more robust against FGSM (Goodfellow et al., 2015) and PGD attacks (Madry et al., 2018) under various levels of perturbation budgets than standard DNNs trained with CE loss in MNIST dataset (LeCun et al., 1998). In our experiments, we also observed that constraining NCMI on benign samples alongside TRADES (Zhang et al., 2019) enhances DNNs' robustness on CIFAR-{10, 100} datasets, both with and without additional synthetic data. We mostly followed the experimental setting of Yang et al. (2023), where we additionally sample 8 instances per class (64 in original CIFAR-10) in each mini-batch and use such sampled instances to update centroid of each benign cluster per mini-batch with a WA factor $\tau_Q = 0.999$ in original CIFAR-100. In experiments on CIFAR datasets with an additional 1 million synthetic images, we increased the sample size for centroid updates in each mini-batch by a factor of 4, aligning with the batch size, which was 4 times compared to the original CIFAR datasets. We also adjusted the WA factor $\tau_Q$ to 0.99.

Indicated in Table 11, constraining NCMI over benign samples also enhances adversarial robustness of a DNN in comparison with models trained with standard TRADES as objective function. To interpret the effectiveness of constraining NCMI over benign samples, we argue that the output probability distributions of a DNN become more compact within each benign sample cluster and better separated among different clusters comparing with the counterpart trained without constraining on benign NCMI. This approach makes it harder for adversarial attacks to create perturbations that would shift a sample into a cluster with a different label in output probability space, thereby improving robust performance. However, we found that constraining NCMI on benign samples cannot outperform directly constraining $\hat{I}(\epsilon)$ in terms of robust accuracy in most cases. We attribute this phenomenon as constraining $\hat{I}(\epsilon)$ takes direct effect on adversarial output probability distribution space rather than benign one, which enhances the robust performance even more.

