# OpenReview forum: "Enhancing Adversarial Robustness Through Robust Information Quantities"
_ICLR.cc/2025/Conference — ICLR 2025 Conference Withdrawn Submission_

### Official Review · Reviewer_StKw · 2024-10-29

**Soundness:** 3
**Presentation:** 3
**Contribution:** 3
**Rating:** 6
**Confidence:** 3

**Summary:**

This paper introduces several new quantities to improve the model robustness, which can also serve as measures of robust performance. The paper provides theoretical proofs for the proposed theorems and uses experiments across multiple datasets to demonstrate the effectiveness for the proposed method against various types of attacks.

**Strengths:**

1. This paper is well-written and easy to follow.
2. The authors provide detailed proofs for the theorems proposed in this paper.
3. Experiments are conducted on multiple datasets, and the results outperform baseline methods in most scenarios.

**Weaknesses:**

1. The primary concern with this paper is its novelty. It appears to apply conditional mutual information from paper [1] to the field of model robustness, so a clearer explanation of the paper's novel contributions would be beneficial.
2. The paper provides the model robustness results against white-box adversarial attacks. It would strengthen the evaluation if the black-box attacks also be included as a metric to assess the robustness of a model.
3. Appendix F presents the adversarial results against different perturbation budget, it would be better if TRADES and MART are also included as baseline models in this figure.
4. How are the default settings for hyper-parameters $\alpha$ and $\beta$ determined?

[1] En-Hui Yang, Shayan Mohajer Hamidi, Linfeng Ye, Renhao Tan, and Beverly Yang. Conditional mutual information constrained deep learning for classification. arXiv preprint arXiv:2309.09123, 2023.

**Questions:**

Please refer to the weaknesses section. Additionally, would it be possible to move the visualization and pseudo-code from the appendix to the main text.

---

> ### Author Response · Authors · 2024-11-26
> **Reply to Reviewer StKw**
>
> Dear Reviewer StKw,
>
> We deeply appreciate your detailed and positive reviews to our work. Please find our answers below in response to your concerns.
>
> ## Replies to Weaknesses
>
> ### 1. It appears to apply conditional mutual information from paper [1] to the field of model robustness, so a clearer explanation of the paper's novel contributions would be beneficial.
>
> Clearly there is misunderstanding here. This paper is very novel. There is huge difference between robust CMI, robust separation, and robust NCMI proposed in this paper, and their clean counterparts CMI, separation, and NCMI in [1].
>
> The concepts of CMI, separation, and NCMI in [1] are defined for clean, original data samples. Since original data samples are given, these information quantities can be computed directly via analytical formulas.
>
> In contrast, the concepts of robust CMI, robust separation, and robust NCMI we proposed in this paper are meant to measure robust mapping structural properties of a DNN against all possible attacks on the original data samples. Since we need to handle all possible attacks on original data samples under a given attack budget, the attacked inputs to the DNN are actually unknown. Each of these proposed robust information quantities is defined through an optimization problem (see Equations (25) to (27) on Page 6). Given any original dataset, there is no simple analytical formula for computing each of them. One has to develop optimization algorithms for computing them numerically. This is why we need Theorem 2 on Pages 6 and 7.
>
> For people who know information theory and lossless/lossy compression, here is the analogy. The difference between our proposed information quantities in this paper and the information quantities in [1] is similar to that between Shannon entropy and rate distortion function or that between lossless compression and lossy compression.
>
> ### 2. It would strengthen the evaluation if the black-box attacks also be included as a metric to assess the robustness of a model.
>
> Thanks for your valuable advice. Please kindly check our experimental results presented in Appendix I.3 for detail. For your convenience, we presented the results of our approach and baseline methods on original CIFAR-10 dataset against Square attack when the underlying DNN model is PreAct-ResNet18 here, where models trained by our approach are more robust than the corresponding baseline methods.
>
> | **Method** | **Clean Acc.** | **Robust Acc.** |
> |:----------:|:--------------:|:---------------:|
> | Vanilla AT | 79.35          | 55.00           |
> | +Ours      | 81.43          | 56.87           |
> | TRADES     | 82.21          | 56.27           |
> | +Ours      | 82.26          | 57.01           |
> | MART       | 79.81          | 54.84           |
> | +Ours      | 79.40          | 55.71           |
> | TRADES-AWP | 81.52          | 56.03           |
> | +Ours      | 81.73          | 56.71           |
>
> ### 3. Appendix F presents the adversarial results against different perturbation budget, it would be better if TRADES and MART are also included as baseline models in this figure.
>
> Thanks for your feedback. We have updated the results in Appendix F, and included TRADES and MART as the baseline models alongside Vanilla AT. In our results, our approach is more robust than plain methods of TRADES and MART against selected perturbation budgets, where we achieve maximum robust accuracy gain of 0.84\% and 1.08\% for TRADES and MART, respectively. Please kindly check that section.
>
> ### 4. How are the default settings for hyper-parameters $\alpha$ and $\beta$ determined?
>
> Thank you for your question. In our setup, we firstly determined the value of $\alpha$ for each of our selected DNN model by choosing the optimal $\alpha=\alpha^*$ which yields the best robust performance when $\beta$ is fixed at 0. Subsequently, we set $\alpha = \alpha^*$ and search for the best $\beta = \beta^*$ that achieves the best robust performance. Eventually, we treat $(\alpha^*, \beta^*)$ as our optimal hyper-parameter setting. Note that both values of $\alpha$ and $\beta$ are selected from the set [0.001, 0.01, 0.1, 1, 2, 5].
>
> ## Replies to Questions
>
> ### 1. Additionally, would it be possible to move the visualization and pseudo-code from the appendix to the main text.
>
> Thanks for your suggestion. It is our original intention to include both sections of visualization and pseudo-code in the main body. However, due to the  requirement of ICLR conference, doing so may make our paper exceed the page limit. Hopefully you can understand this.
>
>
> We thank you again for your time and effort to review our paper.  If our responses above address all your concerns, could you kindly increase your score to a higher level.

---

> ### Comment · Reviewer_StKw · 2024-12-01
>
> Thank you for the response. The authors clarified the differences between normal CMI, separation, NCMI, and their proposed robust quantities. The evaluation against black-box attacks demonstrates the method's performance. The response addressed my concerns, and I will maintain my positive rating.

---

### Official Review · Reviewer_uHbR · 2024-10-30

**Soundness:** 3
**Presentation:** 3
**Contribution:** 2
**Rating:** 6
**Confidence:** 4

**Summary:**

This paper uses CMI and normalized CMI (NCMI) to improve an established defense (adversarial training) against adversarial attacks. The authors take CMI, NCMI and robust separation as three measurements to quantify the robustness.
Extensive experiments are conducted to verify the effectiveness of the proposed method.

**Strengths:**

- This paper provides both theoretical and experimental results to support their statements.
- It is easy to follow the presentation of the paper.
- Incorporating CMI and NCMI into adversarial training.

**Weaknesses:**

- The experimental improvement does not seem significant in Table 1 and 2. In a few cases, it even decreases more on clean accuracy compared to the increase in robust accuracy.
- This paper does not include enough baselines, only TRADES and MART are included. There are many papers that use MI to improve adversarial robustness, such as HBaR[1], VIB[2], and IB-RAR[3].
- Need more concrete ablation study, such as the different values of $\alpha$ and $\beta$ in equation(40).

[1] Revisiting Hilbert-Schmidt Information Bottleneck for Adversarial Robustness

[2] Deep variational information bottleneck

[3] IB-RAR: Information Bottleneck as Regularizer for Adversarial Robustness

**Questions:**

- How to choose the $\lambda$ value in equation(33)? Or how to choose the $\alpha$ and $\beta$ values in equation(40)?

- Why is the code for fixing random seeds in the GitHub page commented out? Given that the improvement is minor in Table 1 and 2, I would expect to see the variance or error bars (maybe only a part of the experiments due to high computation costs) to see if there are overlaps among error bars.

- What is the difference between the proposed CMI, NCMI, and CMI, NCMI in [4]?

I will increase my rating if my concerns are properly addressed.


[4] Conditional mutual information constrained deep learning for classification.

---

> ### Author Response · Authors · 2024-11-26
> **Reply to Reviewer uHbR (1/3)**
>
> Dear Reviewer uHbR,
>
> Thank you for your detailed and thoughtful reviews. Please find our answers below in response to your concerns.
>
> ## Replies to Weaknesses (Part 1)
>
> ### 1. The experimental improvement does not seem significant in Table 1 and 2. In a few cases, it even decreases more on clean accuracy compared to the increase in robust accuracy.
>
> Thank you for your comment. While we acknowledge that our improvements are not always significant, our key messages and contributions are as follows:
>
> First, our improvements are consistent across models, datasets, and different benchmark methods. Given that we are not motivated to improve any specific method, this is no small feat. Our method is motivated by investigating robust information geometrical properties of DNNs, generic, orthogonal to state-of-the-art benchmark methods, and of plug-and-play nature.
>
> Second, since the capacity of our selected WRN models are quite large, it would be more difficult to improve the performances of such models than models with smaller capacity. In addition, the improvement of our method is more significant when the number of classes in a given dataset increases. This is because that there is more structural information encapsulated in DNN's output probability distribution space when the number of classes of a dataset is high, so there will be more room for the robust NCMI to be minimized. In the case of CIFAR-10, the number of classes is small; so the room for us to improve is also small.
>
> Third, more importantly, our major contributions lie in the introduction of robust structural properties of DNNs as defined by our new robust information quantities: robust conditional mutual information (CMI), robust separation, and robust normalized CMI (NCMI). These are new robust metrics different from robust error rate. The main purpose of our paper is to show that by constraining and controlling these structural quantities during the training process, we can not only improve the normal robust accuracy consistently across models and datasets, but also achieve other fringe benefits such as those reported in Appendix F in terms of robustness against other variations.
>
> ### 2. This paper does not include enough baselines, only TRADES and MART are included. There are many papers that use MI to improve adversarial robustness, such as HBaR, VIB, and IB-RAR.
>
> Thank you for mentioning this. We have selected some of the above methods to compare with ours. Please kindly find our experimental results in Appendix I.4 for detail. For your convenience, we also present the results here. Note that we report results for the best checkpoints obtained for each method on original CIFAR-10 dataset when the underlying DNN architecture is ResNet18.
>
> | **Method** | **Clean**      | **PGD**        | **C\&W$_\infty$** | **AA**         |
> |:----------:|:--------------:|:--------------:|:---------------:|:--------------:|
> | Vanilla AT | 83.65          | 51.64          | 50.66           | 48.10          |
> | +HBaR      | **83.96**   | 52.49          | 50.70           | 48.55          |
> | +IB-RAR    | 83.71          | 52.53          | 50.73           | 48.63          |
> | +Ours      | 83.04          | **52.80** | **51.06**  | **48.79** |
> | TRADES     | 83.36          | 51.93          | 50.40           | 49.00          |
> | +HBaR      | 83.39          | 52.49          | **51.09**  | 49.30          |
> | +IB-RAR    | **83.44** | 52.58          | 50.67           | 49.56          |
> | +Ours      | 82.75          | **52.87** | 51.05           | **50.09** |

---

> > ### Author Response · Authors · 2024-11-26
> > **Reply to Reviewer uHbR (2/3)**
> >
> > ## Replies to Weakness (Part 2)
> > ### 3. Need more concrete ablation study, such as the different values of $\alpha$ and $\beta$ in equation(40).
> >
> > Thank you for your valuable advice. Please kindly find our ablation study in Appendix H for detail. For your convenience, we also provided results of adopting various values of our hyper-parameters $\alpha$ and $\beta$ on original CIFAR-10 dataset when the underlying DNN model is PreAct-ResNet18 here.
> >
> > | **Hyper-Param**                  | **Clean**      | **PGD**        | **C\&W$_\infty$** | **AA**         |
> > |:--------------------------------:|:--------------:|:--------------:|:---------------:|:--------------:|
> > | $\alpha=0$, $\beta=0$ (Baseline) | 79.35          | 52.51          | 50.76           | 48.70          |
> > | $\alpha=0.001$, $\beta=0$        | 82.25          | 52.56          | 50.93           | 48.73          |
> > | $\alpha=0.01$, $\beta=0$         | 81.95          | 52.27          | 50.86           | 48.79          |
> > | $\alpha=0.1$, $\beta=0$          | **82.63** | 52.65          | 50.79           | 48.82          |
> > | $\alpha=1$, $\beta=0$            | 81.32          | **52.83** | **51.99**  | **49.39** |
> > | $\alpha=2$, $\beta=0$            | 80.37          | 52.61          | 51.51           | 49.01          |
> > | $\alpha=5$, $\beta=0$            | 79.96          | 52.78          | 50.94           | 48.99          |
> > | $\alpha=0$, $\beta=0.001$        | 79.49          | 52.60          | 50.79           | 48.72          |
> > | $\alpha=0$, $\beta=0.01$         | 79.43          | **52.79** | 50.80           | 48.74          |
> > | $\alpha=0$, $\beta=0.1$          | **81.72** | 52.54          | 51.38           | 48.79          |
> > | $\alpha=0$, $\beta=1$            | 80.93          | 52.76          | 52.12           | 49.34          |
> > | $\alpha=0$, $\beta=2$            | 80.72          | 52.59          | **52.18**  | **49.35** |
> > | $\alpha=0$, $\beta=5$            | 79.72          | 52.61          | 51.94           | 49.30          |
> > | $\alpha=1$, $\beta=2$ (Best)     | 81.43          | **53.04** | **52.23**  | **49.47** |
> >
> > In general, solely enabling each component can improve the robustness of the DNN model, and enabling both components simultaneously can further enhance the robust performance.

---

> > > ### Author Response · Authors · 2024-11-26
> > > **Reply to Reviewer uHbR (3/3)**
> > >
> > > ## Replies to Questions
> > >
> > > ### 1. How to choose the $\lambda$ value in equation(33)? Or how to choose the $\alpha$ and $\beta$ values in equation(40)?
> > >
> > > Thank you for your question. In our setup, we firstly determined the value of $\alpha$ for each of our selected DNN model by choosing the optimal $\alpha=\alpha^*$ which yields the best robust performance when $\beta$ is fixed at 0. Subsequently, we set $\alpha = \alpha^*$ and search for the best $\beta = \beta^*$ that achieves the best robust performance. Eventually, we treat $(\alpha^*, \beta^*)$ as our optimal hyper-parameter setting. Note that both values of $\alpha$ and $\beta$ are selected from the set [0.001, 0.01, 0.1, 1, 2, 5].
> > >
> > > ### 2. Given that the improvement is minor in Table 1 and 2, I would expect to see the variance or error bars (maybe only a part of the experiments due to high computation costs) to see if there are overlaps among error bars.
> > >
> > > Thanks for raising this up. Please kindly check Appendix J where we reported the variance of our experimental results.
> > >
> > > ### 3. What is the difference between the proposed CMI, NCMI, and CMI, NCMI in [4]?
> > >
> > > We did not propose CMI and NCMI in this paper. Instead, we proposed robust CMI, robust separation, and robust NCMI. There is huge difference between robust CMI, robust separation, and robust NCMI, and their clean counterparts CMI, separation, and NCMI.
> > >
> > > The concepts of CMI, separation, and NCMI in [4] are defined for clean, original data samples. Since original data samples are given, these information quantities can be computed directly via analytical formulas.
> > >
> > > In contrast, the concepts of robust CMI, robust separation, and robust NCMI we proposed in this paper are meant to measure robust mapping structural properties of a DNN against all possible attacks on the original data samples. Since we need to handle all possible attacks on original data samples under a given attack budget, the attacked inputs to the DNN are actually unknown. Each of these proposed robust information quantities is defined through an optimization problem (see Equations (25) to (27) on Page 6). Given any original dataset, there is no simple analytical formula for computing each of them. One has to develop optimization algorithms for computing them numerically. This is why we need Theorem 2 on Pages 6 and 7.
> > >
> > > For people who know information theory and lossless/lossy compression, here is the analogy. The difference between our proposed information quantities in this paper and the information quantities in [4] is similar to that between Shannon entropy and rate distortion function or that between lossless compression and lossy compression.
> > >
> > >
> > > We thank you again for your insightful reviews.  If our responses above address all your concerns, could you kindly increase your score to a higher level.

---

> > > > ### Comment · Reviewer_uHbR · 2024-11-27
> > > >
> > > > Thanks for the authors' reply. Most of my concerns are addressed. I will increase my rating. I also encourage the authors to emphasize their motivation more in the introduction and include more complete experiments. For example, using the experimental setting in Table 2 for evaluation on Tiny-ImageNet and evaluating HBaR and IB-RAR as more proper baselines in Table 2.

---

### Official Review · Reviewer_nVqT · 2024-11-01

**Soundness:** 2
**Presentation:** 3
**Contribution:** 3
**Rating:** 5
**Confidence:** 4

**Summary:**

Inspired by the information geometry analyzed in the previous work, this paper proposes three advanced metrics which are robust CMI (Conditional Mutual Information, CMI), robust separation and robust NCMI (Normalized CMI, NCMI), where they can be incorporated into adversarial training methods for further enhancing the model’s robustness.

**Strengths:**

1. The mathematical analysis is adequate for enhancing the reliability. Detailed theoretical introductions in the paper and the supplementary materials demonstrate the feasibility of information theory in the adversarial robustness.

2. The consideration of the perturbation in the worst case for learning general robust features makes a rational extension for these metrics.

**Weaknesses:**

1. The Introduction lacks the summary about the main problem mentioned in the third paragraph of this section. Authors state that data which are near the decision boundary are in general more susceptible, and then intrdouce the definition of proposed metrics. However, how does the proposed defense solve this problem? Providing additional analyses about it may be better.

2. The evaluations with and without synthetic data are somewhat confusing in terms of the used model’s capacity. In Table 2, why the smaller model WRN-28-10 was used for the larger dataset containing synthetic data instead of using the larger WRN-34-10? Please provide explanations for such an inconsistency.

3. It seems that ablation studies for hyper-parameters are not shown. Adding comprehensive ablation studies may be better for verifying the effectiveness of each module. Additionally, the authors can ablate the individual components of the proposed method (e.g. robust CMI vs robust separation) to demonstrate the contribution of each.

4. Cross-norm evaluations may be more convincing for the generalization ability of the proposed method. There are various evaluations about the robustness under L∞-norm, while the L2-norm attacks also deserve to be evaluated. The authors can include evaluations against specific L2-norm attacks (e.g. PGD-L2, C&W-L2) in their main results tables, in addition to the existing L$_\infty$-norm evaluations.

**Questions:**

While authors evaluate the robustness of the checkpoint with the highest validation accuracy, I wonder if evaluations from the last checkpoint can also be presented here. Besides, the authors can include a comparison of robustness results between the checkpoint with highest validation accuracy and the last checkpoint. This comparison could reflect the potential overfitting or the stability of the robustness gains over the course of training.

---

> ### Author Response · Authors · 2024-11-26
> **Reply to Reviewer nVqT (1/3)**
>
> Dear Reviewer nVqT,
>
> We are sincerely thankful to your insightful comments and reviews. Please find our responses to your concerns below.
>
> ## Replies to Weaknesses (Part 1)
>
> ### 1. Authors state that data which are near the decision boundary are in general more susceptible, and then introduce the definition of proposed metrics. However, how does the proposed defense solve this problem? Providing additional analyses about it may be better.
>
> Thank you for pointing this out. In response to this question, please kindly refer to Appendix G, where we visualized the simplex of DNN probability outputs. The subplot titled `$\alpha=0$, $\beta=0$' in Fig. 5 represents the output of baseline method (Vanilla AT), where the rest of subplots in Fig. 5 and 6 are visualizations of models trained with our approach. When we fix the hyper-parameter $\alpha=1$, and $\beta$ is set to 2 or 5, the overlapping areas in the simplex among different classes are smaller than that of the baseline method, as we enforce (1) data points to be more concentrated around the centroid of corresponding cluster and (2) different clusters to be more separated with each other. In other words, less data points are near the decision boundaries and susceptible against adversarial attacks under our proposed framework comparing with baseline, and enhances the robustness of a model in general.
>
> ### 2. In Table 2, why the smaller model WRN-28-10 was used for the larger dataset containing synthetic data instead of using the larger WRN-34-10? Please provide explanations for such an inconsistency.
>
> Thanks for bringing this out, and we would love to clear this confusion for you. In the original literature of all our selected baseline methods, they tested their methods with WRN-34-10 on original dataset(s), while prior literature utilizing additional synthetic data to improve robustness mostly adopted WRN-28-10 to conduct their experiments (e.g., [1-3]). We trying to make a fair comparison with these methods by following their settings. Another reason that training a smaller WRN-28-10 model on larger datasets is in consideration of training time, as training on such datasets is generally more computationally expensive.
>
> [1] Rebuffi, S., Gowal, S., Calian, D. A., Stimberg, F., Wiles, O., and Mann, T. A. Fixing data augmentation to improve adversarial robustness.
>
> [2] Gowal, S., Rebuffi, S., Wiles, O., Stimberg, F., Calian, D. A., and Mann, T. A. Improving robustness using generated data.
>
> [3] Wang, Z., Pang, T., Du, C., Lin, M., Liu, W., and Yan, S. Better diffusion models further improve adversarial training.

---

> > ### Author Response · Authors · 2024-11-26
> > **Reply to Reviewer nVqT (2/3)**
> >
> > ## Replies to Weakness (Part 2)
> > ### 3. It seems that ablation studies for hyper-parameters are not shown. Adding comprehensive ablation studies may be better for verifying the effectiveness of each module. Additionally, the authors can ablate the individual components of the proposed method (e.g. robust CMI vs robust separation) to demonstrate the contribution of each.
> >
> > Thanks for your valuable advice. We have added a section of ablation study in Appendix H. Please kindly refer to that section for detail. We also provided results of adopting various values of our hyper-parameters $\alpha$ and $\beta$ on original CIFAR-10 dataset when the DNN model is PreAct-ResNet18 here for your reference.
> >
> > | **Hyper-Param**                  | **Clean**      | **PGD**        | **C\&W$_\infty$** | **AA**         |
> > |:--------------------------------:|:--------------:|:--------------:|:---------------:|:--------------:|
> > | $\alpha=0$, $\beta=0$ (Baseline) | 79.35          | 52.51          | 50.76           | 48.70          |
> > | $\alpha=0.001$, $\beta=0$        | 82.25          | 52.56          | 50.93           | 48.73          |
> > | $\alpha=0.01$, $\beta=0$         | 81.95          | 52.27          | 50.86           | 48.79          |
> > | $\alpha=0.1$, $\beta=0$          | **82.63** | 52.65          | 50.79           | 48.82          |
> > | $\alpha=1$, $\beta=0$            | 81.32          | **52.83** | **51.99**  | **49.39** |
> > | $\alpha=2$, $\beta=0$            | 80.37          | 52.61          | 51.51           | 49.01          |
> > | $\alpha=5$, $\beta=0$            | 79.96          | 52.78          | 50.94           | 48.99          |
> > | $\alpha=0$, $\beta=0.001$        | 79.49          | 52.60          | 50.79           | 48.72          |
> > | $\alpha=0$, $\beta=0.01$         | 79.43          | **52.79** | 50.80           | 48.74          |
> > | $\alpha=0$, $\beta=0.1$          | **81.72** | 52.54          | 51.38           | 48.79          |
> > | $\alpha=0$, $\beta=1$            | 80.93          | 52.76          | 52.12           | 49.34          |
> > | $\alpha=0$, $\beta=2$            | 80.72          | 52.59          | **52.18**  | **49.35** |
> > | $\alpha=0$, $\beta=5$            | 79.72          | 52.61          | 51.94           | 49.30          |
> > | $\alpha=1$, $\beta=2$ (Best)     | 81.43          | **53.04** | **52.23**  | **49.47** |
> >
> > In general, solely enabling each component can improve the robustness of the DNN model, and enabling both components simultaneously can further enhance the robust performance.
> >
> > ### 4. The authors can include evaluations against specific L2-norm attacks (e.g. PGD-L2, C\&W-L2) in their main results tables, in addition to the existing L$_\infty$-norm evaluations.
> >
> > Thank you for your comment. We have evaluated the performance of our models reported in Table 2 (trained with data perturbed in $L_\infty$-norm) against L2-norm attacks in Appendix I.1. Please kindly refer to that section for detail. We also provided the results on original CIFAR-10 dataset here for your reference, where models trained with our approach exhibit better robustness against PGD-L2 and C\&2-L2 attacks in comparison with corresponding baseline methods in most cases.
> >
> > **Results for PreAct-ResNet18**:
> > | **Method** | **PGD-L2** | **C\&W-L2** |
> > |:----------:|:----------:|:---------:|
> > | Vanilla AT | 62.22      | 60.19     |
> > | +Ours      | 63.85      | 62.13     |
> > | TRADES     | 63.03      | 61.47     |
> > | +Ours      | 63.53      | 61.97     |
> > | MART       | 63.75      | 60.29     |
> > | +Ours      | 63.60      | 61.06     |
> > | TRADES-AWP | 63.23      | 61.19     |
> > | +Ours      | 63.76      | 61.45     |
> >
> > **Results for WRN-34-10**:
> > | **Method** | **PGD-L2** | **C\&W-L2** |
> > |:----------:|:----------:|:---------:|
> > | Vanilla AT | 63.61      | 63.07     |
> > | +Ours      | 65.27     | 63.23     |
> > | TRADES     | 63.48      | 62.45     |
> > | +Ours      | 63.93      | 62.79     |
> > | MART       | 64.93      | 62.49     |
> > | +Ours      | 64.73      | 63.09     |
> > | TRADES-AWP | 64.56      | 62.92     |
> > | +Ours      | 64.81      | 63.91     |

---

> > > ### Author Response · Authors · 2024-11-26
> > > **Replies to Reviewer nVqT (3/3)**
> > >
> > > ## Replies to Questions
> > >
> > > ### 1. The authors can include a comparison of robustness results between the checkpoint with highest validation accuracy and the last checkpoint.
> > >
> > > Thanks for mentioning this. Comparison between best and last checkpoints are presented in Appendix I.2. Please kindly check this section for detail. For your convenience, we present our results of last checkpoints on original CIFAR-10 here, where our approach exhibits better robust performance in the last model checkpoint comparing with each baseline counterpart.
> > >
> > > **Results of PreAct-ResNet18**:
> > > | **Method** | **Clean** |  **PGD**  | **C\&W$_\infty$** |   **AA**  |
> > > |------------|:---------:|:---------:|:-----------------:|:---------:|
> > > | Vanilla AT |   78.89   |   43.18   |       44.08       |   41.53   |
> > > | +Ours      | **80.66** | **46.63** |     **47.32**     | **43.98** |
> > > | TRADES     |   81.57   |   50.47   |       49.26       |   47.40   |
> > > | +Ours      | **82.20** | **50.77** |     **49.67**     | **48.19** |
> > > | MART       | **79.58** |   47.26   |       45.78       |   42.71   |
> > > | +Ours      |   79.22   | **47.52** |     **46.04**     | **43.66** |
> > > | TRADES-AWP |   81.22   |   53.14   |       50.97       |   49.71   |
> > > | +Ours      | **81.40** | **53.71** |     **51.56**     | **50.64** |
> > >
> > > **Results of WRN-34-10**:
> > > | **Method** | **Clean** |  **PGD**  | **C\&W$_\infty$** |   **AA**  |
> > > |------------|:---------:|:---------:|:-----------------:|:---------:|
> > > | Vanilla AT |   84.05   |   46.93   |       47.73       |   45.38   |
> > > | +Ours      | **84.48** | **47.16** |     **48.20**     | **46.15** |
> > > | TRADES     | **85.44** |   48.98   |       50.06       |   47.82   |
> > > | +Ours      |   85.08   | **49.14** |     **50.21**     | **48.22** |
> > > | MART       | **83.63** |   48.06   |       47.71       |   45.04   |
> > > | +Ours      |   83.58   |   48.05   |     **47.90**     | **45.46** |
> > > | TRADES-AWP | **84.17** |   50.52   |       51.06       |   48.94   |
> > > | +Ours      |   84.05   | **50.84** |     **51.23**     | **49.34** |
> > >
> > > We thank you again for your thoughtful reviews. If our responses above address all your concerns, could you kindly increase your score to a higher level.

---

> > > > ### Comment · Reviewer_nVqT · 2024-11-27
> > > > **Response to authors' rebuttal**
> > > >
> > > > Thank authors for their responses and explanations. For the first concern, authors provide the analysis, but the motivation and solutions for how to solve the found problems need to be clearly and concisely stated in Introduction. Besides, for the second concern, a closely related reason is still missing, there is no direct correlation between making fair comparisons and using a smaller model for larger data, authors could reproduce the baselines using the corresponding models to make fair comparisons, like a wideresnet is generally used for Tiny-ImageNet and a resnet is used for CIFAR-10. In addition, according to the results of last checkpoint, the proposed method brings marginal improvements, especially for the stronger network. Thus, I'm sorry I'll keep the original rating score.

---

> ### Author Response · Authors · 2024-12-01
> **Reply to Reviewer nVqT**
>
> Dear Reviewer nVqT,
>
> We sincerely appreciate you for your time and effort as well as your additional feedback. Please kindly find our further responses below.
>
> As requested, we have revised our Introduction section to address your first concern (Please see texts written in blue in Introduction). For your second concern, we have conducted experiments on CIFAR-{10,100} datasets plus additional 1M synthetic data with WRN-34-10. Due to the limited time we have, we only tested our method combining with vanilla AT and TRADES, where the conclusion is similar to that when the underlying DNN model is WRN-28-10 as reported in Table 2. Our approach consistently outperform each baseline method. Please find our results below:
>
> **Results of WRN-34-10 on CIFAR-10 with 1M Synthetic Data**:
>
> | **Method** | **Clean**      | **PGD**        | **C\&W$_\infty$** | **AA**         |
> |:----------:|:--------------:|:--------------:|:---------------:|:--------------:|
> | Vanilla AT | **93.06**          | 64.33          | 64.25           | 61.97          |
> | +Ours      | 92.73         | **64.82** | **64.96**  | **62.48** |
> | TRADES     | **90.79**          | 65.37          | 64.74           | 63.32         |
> | +Ours      | 90.63          | **65.96** | **65.12**           | **63.93** |
>
> **Results of WRN-34-10 on CIFAR-100 with 1M Synthetic Data**:
>
> | **Method** | **Clean**      | **PGD**        | **C\&W$_\infty$** | **AA**         |
> |:----------:|:--------------:|:--------------:|:---------------:|:--------------:|
> | Vanilla AT | 71.37         | 35.87          | 37.16           | 34.61          |
> | +Ours      | 71.37         | **36.80** | **37.89**  | **35.22** |
> | TRADES     | 68.02          | 39.34          | 37.43           | 36.16         |
> | +Ours      | **68.08**          | **39.82** | **37.53**           | **36.44** |
>
> For the results of the last checkpoint, we additionally evaluate on original CIFAR-100 and present the results below. We argue that the improvements are more significant than those on original CIFAR-10. This is because that there is more structural information encapsulated in DNN's output probability distribution space when the number of classes of a dataset increases, so there will be more room for the robust NCMI to be minimized. In the case of CIFAR-10, the number of classes is small; so the room for us to improve is also small. As the number of classes increases to 100 for CIFAR-100, the room for us to improve is larger. We present the results for last checkpoint on original CIFAR-100 below for your reference.
>
> **Results of PreAct-ResNet18**:
>
> | **Method** | **Clean** | **PGD** | **C\&W$_\infty$** | **AA** |
> |------------|-----------|---------|--------|--------|
> | Vanilla AT | 55.55     | 19.91   | 20.45  | 18.73  |
> | +Ours      | **55.72**     | **25.86**   | **23.68**  | **21.66**  |
> | TRADES     | 55.16     | 26.69   | 23.87  | 22.88  |
> | +Ours      | **57.55**     | **28.34**   | **24.97**  | **23.92**  |
> | MART       | **53.15**     | 22.61   | 21.12  | 19.71  |
> | +Ours      | 52.68     | **23.71**   | **22.34**  | **20.78**  |
> | TRADES-AWP | **55.81**     | 28.38   | 24.99  | 24.09  |
> | +Ours      | 55.40     | **28.96**   | **25.76**  | **24.88**  |
>
> **Results of WRN-34-10**:
>
> | **Method** | **Clean** | **PGD** | **C\&W$_\infty$** | **AA** |
> |------------|-----------|---------|--------|--------|
> | Vanilla AT | **57.35**     | 23.19   | 23.08  | 21.73  |
> | +Ours      | 57.09     | **24.19**   | **23.81**  | **22.34**  |
> | TRADES     | 56.02     | 25.30   | 24.31  | 23.29  |
> | +Ours      | **57.35**     | **26.19**   | **25.77**  | **24.29**  |
> | MART       | **56.06**     | 23.28   | 22.73  | 21.03  |
> | +Ours      | 55.75     | **24.18**   | **23.14**  | **22.06**  |
> | TRADES-AWP | 58.11     | 28.92   | 27.24  | 26.25  |
> | +Ours      | **58.56**     | **30.22**   | **28.36**  | **27.09**  |
>
> We genuinely hope our responses have further alleviated your concerns. If that is the case, we wonder if you could kindly reconsider about giving us a higher rating?

---

### Official Review · Reviewer_dKSG · 2024-11-05

**Soundness:** 3
**Presentation:** 3
**Contribution:** 2
**Rating:** 6
**Confidence:** 4

**Summary:**

The paper develops a new regularizer for adversarial training (AT) by extending the work of Yang et al. (2023) and introduces three new information-based metrics: robust conditional mutual information (CMI), robust separation, and robust normalized CMI (NCMI). The authors theoretically and empirically demonstrate that adversarial robustness is inversely proportional to adversarial NCMI. Based on this insight, they design an objective function that combines the AT loss with an NCMI-based regularization term. To optimize this objective, they propose a relaxed alternating algorithm.

**Strengths:**

- The paper is technically sound, with the objective function being theoretically justified and proven.
- It shows improvement to white-box attacks.
- The paper is well-written and easy to follow.
- Experiments show improvement across the board.

**Weaknesses:**

- While the improvement in results is consistent, it is often quite small. For example, the best AA accuracy for CIFAR-10 with WRN is improved by only about 0.25%.
- Given the iterative nature of the algorithm, it would be useful to know the time complexity of training relative to baseline methods.

**Questions:**

1. How does the training time compare to, say, TRADES?
2. Besides training time, are there other limitations or weaknesses of the method? For instance, how does it handle class imbalance?
3. Is the drop in natural accuracy expected (theoretically)?

Additional questions (not directly related to the paper but relevant to adversarial robustness):

4. Adversarial training (AT) is often limited to very small images, and the white-box accuracy is often too low for reliable deployment. Do the authors have any ideas on how to scale AT effectively?

5. Methods based on stable diffusion have shown significantly improved performance in adversarial robustness. Could the authors comment on how their approach compares to these methods, and whether diffusion-based techniques might complement or enhance their framework?

---

> ### Author Response · Authors · 2024-11-26
> **Reply to Reviewer dKSG (1/2)**
>
> Dear Reviewer dKSG,
>
> We genuinely thank your time and effort as well as your positive comments to our paper. Below please find our point-to-point replies to your mentioned weakness and questions.
>
> ## Replies to Weaknesses
> ### 1. While the improvement in results is consistent, it is often quite small. For example, the best AA accuracy for CIFAR-10 with WRN is improved by only about 0.25\%.
>
> We acknowledged this point. However, this should be judged from two complementary perspectives.
>
> First, since the capacity of our selected WRN models are quite large, it would be more difficult to improve the performances of such models than models with smaller capacity. In addition, the improvement of our method is more significant when the number of classes in a given dataset increases. This is because that there is more structural information encapsulated in DNN's output probability distribution space when the number of classes of a dataset is high, so there will be more room for the robust NCMI to be minimized. In the case of CIFAR-10, the number of classes is small; so the room for us to improve is also small.
>
> Second, more importantly, our major contributions lie in the introduction of robust structural properties of DNNs as defined by our new robust information quantities: robust conditional mutual information (CMI), robust separation, and robust normalized CMI (NCMI). These are new robust metrics different from robust error rate. The main purpose of our paper is to show that by constraining and controlling these structural quantities during the training process, we can not only improve the normal robust accuracy consistently across models and datasets, but also achieve other fringe benefits such as those reported in Appendix F in terms of robustness against other variations.
>
> ### 2. Given the iterative nature of the algorithm, it would be useful to know the time complexity of training relative to baseline methods.
>
> You probably did not check our Appendix E. The time complexity comparison was provided in our Appendix E on Page 18. In comparison with vanilla AT, TRADES, and MART, our Algorithm 1 and Algorithm 2 require roughly 60\% and 120\% more training time in our setup, whereas TRADES-AWP requires roughly 30\% more training time.

---

> > ### Author Response · Authors · 2024-11-26
> > **Reply to Reviewer dKSG (2/2)**
> >
> > ## Replies to Questions
> > ### 1. How does the training time compare to, say, TRADES?
> >
> > Please kindly refer to our Appendix E and our discussion above.
> >
> > ### 2. Besides training time, are there other limitations or weaknesses of the method? For instance, how does it handle class imbalance?
> >
> > Thanks for your question. Besides training time, we are NOT aware of any other limitations. Since we improve the robust structural properties of DNNS, we believe that our method has other fringe benefits as well such as those reported in Appendix F in terms of robustness against other variations. This may improve the performance in the case of imbalanced datasets. However, we have not conducted any experiment to verify this yet, since our compared benchmark adversarial training methods do not discuss the case of imbalanced datasets either. How to address imbalanced datasets from information theoretic or information geometrical perspectives will be left for future research.
> >
> > ### 3. Is the drop in natural accuracy expected (theoretically)?
> >
> > No, we don't think so.  As the priority objective of this paper is enhancing robust accuracy, we put most of our efforts to improve the robustness of the trained DNNs by constraining their adversarial input-output mapping structures, and we did not explicitly constrain natural accuracy in our theoretical part. Nevertheless, according to our experimental results on original CIFAR datasets (see Table 2), we observed that our method achieved higher natural accuracy when the baseline methods were vanilla AT or TRADES in most cases. In contrast, for the remaining experiments, natural accuracy experienced a slight decline in exchange for improved robustness.
> >
> > ### 4. AT is often limited to very small images, and the white-box accuracy is often too low for reliable deployment. Do the authors have any ideas on how to scale AT effectively?
> >
> > Thank you for this excellent question. At this point, we don't have good ideas on how to scale AT effectively. But it is this type of questions that motivates us to investigate the mapping structural properties of DNNs. As shown in Appendix F, DNNs with improved mapping structural properties indeed have better robustness against unknown variations. We believe that constraining and controlling the mapping structures of DNNs during training would be an indirect way to address the above question. The defined robust information quantities are just some of the mapping structure properties of DNNs.
> >
> > ### 5. Methods based on stable diffusion have shown significantly improved performance in adversarial robustness. Could the authors comment on how their approach compares to these methods, and whether diffusion-based techniques might complement or enhance their framework?
> >
> > Thank you for your question. If you are referring to utilizing additional images synthesized by diffusion models and then perturbing such generated images on top of original data for AT, we showed that our method can further boost the robust performance on top of these synthetic data and outperform the baselines (see column '1M Synthetic Data' in Table 2).
> >
> > If you are referring to adopting diffusion models to purify adversarial data before they were fed to a given pretrained DNN (adversarial purification), our method took a different way to improve robustness in comparison with those approaches. Since we did not attempt to purify the adversarial examples, our method can be considered as orthogonal to adversarial purification methods. In order to enhance such diffusion-based methods with our framework, we can treat the purified images as weakly-perturbed adversarial data (noise may not be guaranteed to be completely removed), and then use our method to fine-tune the pretrained model to constrain its input-output mapping structure to further improve robustness.
> >
> >
> > We thank you for your detailed review again. If our responses above address all your concerns, could you kindly increase your score to a higher level.

---

> > > ### Comment · Reviewer_dKSG · 2024-11-28
> > >
> > > Thankyou for your reply (For Q5, I meant purifying adversarial data). Given the amount of increase in accuracy and complexity of the algorithm, I will keep my original positive score.

---

### Note · Authors · 2025-01-29

I have read and agree with the venue's withdrawal policy on behalf of myself and my co-authors.